# Gene Regulatory Network Inference in the Presence of Selection Bias and Latent Confounders

## Abstract

The study of gene regulatory network inference (GRNI), with a focus on uncovering causal relations among genes, holds significant potential to explain fundamental biological processes, such as how cellular identity is established or disrupted in disease. Unfortunately, current methods fail to adequately interpret the widespread phenomena of differential gene expression. The limitation can largely be attributed to the overlook of the selection process (e.g., survival bias), which is ubiquitous and fundamental in biology. Furthermore, recent studies have shown that gene expression is regulated by latent confounders (e.g., non-coding RNAs). Both of which can lead to spurious dependencies, thereby distorting GRNI results. To mitigate these challenges, we propose a novel algorithm, called Gene Regulatory Network Inference in the presence of Selection bias and Latent confounders (GISL). It is designed to uncover the causal structure by leveraging data across multiple distributions obtained via gene perturbation. Surprisingly, we find that the qualitative structure information, selection process, and latent confounders are partially identifiable without any parametric assumption under mild graphical conditions. Experimental results on both synthetic and real-world single-cell gene expression datasets demonstrate the superiority of GISL over existing strong baseline methods.

## 1 Introduction

Gene Regulatory Networks (GRNs), where nodes represent genes and directed edges signify cross-gene causal relations (Levine & Davidson, 2005), playing a pivotal role in understanding the biological processes at the molecular level and disease mechanisms like cancer (Hanahan & Weinberg, 2000). The differential gene expression (Robinson et al., 2010), particularly the variation in the distribution of the same gene across different cell types, is usually interpreted by latent factors, including but not limited to latent confounders, e.g., Non-coding RNAs, environmental stimuli, and cell type composition (Gasch et al., 2000; Statello et al., 2021; Razin & Gavrilov, 2021). However, the distribution changes of unregulated genes in gene perturbation data raise our curiosity to explore the underlying true data generation process. We argue that this is due to the overlook of the selection process, which is ubiquitous and fundamental in cells. Then, both selection processes and latent confounders lead to spurious edges, which severely bias the GRNI, as they do not have causal relations in between. This motivates us to identify selection processes and latent confounders, and to recover regulatory relations from observed dependencies.

Let us start with a toy example in Figure 1 to show the selection process and how it leads to distribution change despite the absence of the causal relation. We assume $X$ and $Y$ are independent following Normal distribution. When applying a simple selection function (e.g. $1.5X + 1.6Y > 3.2$) on them, we can observe the spurious dependence shown in (b). The causal structure is $X \rightarrow S \leftarrow Y$, where the selection variable $S$ is always given. More interesting is that after perturbing $X$, the distribution of $Y$ changes significantly as shown in (c) with the **variations in sample size** (reduced from 5943 to 2601). Then we continue with some interesting phenomena, which inspire us to figure out why. For example, from the Norman dataset (lung carcinoma cell) (Thomas M. et al., 2019), with perturbing gene TP73, a significant distribution change of gene CENPF is observed as shown in Figure 2. However, with prior knowledge, comprehensive libraries collected by Enrichr (Kuleshov et al., 2016) show that there is no functional relation between gene

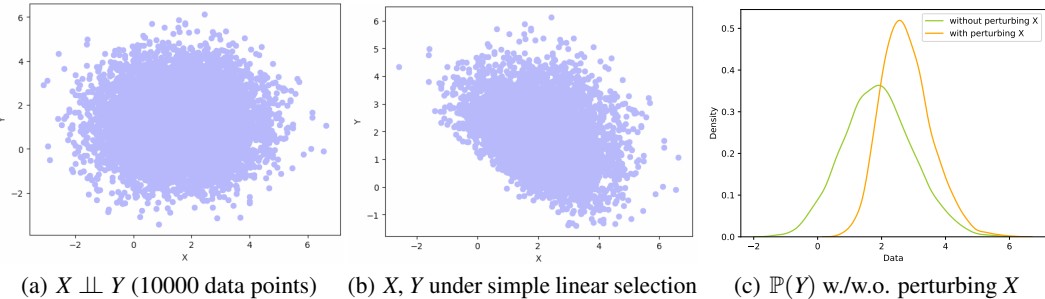

(a) $X \perp\!\!\!\perp Y$ (10000 data points)  (b) $X, Y$ under simple linear selection  (c) $\mathbb{P}(Y)$ w./w.o. perturbing $X$

Figure 1: A toy example to introduce (a & b) the selection process, and (c) how it leads to distribution change despite the absence of the causal relation.

TP73 and CENPF. This interests us in uncovering the causal patterns to explain this phenomenon. Dependencies can be generated in three ways: through causality, latent confounders, or selection bias. The distribution changes of other genes following the perturbation of one gene only occur due to causal mechanisms or selection processes. The simulation in Figure 1, provides a possible explanation of this phenomenon, suggesting gene TP73 and CENPF are under the selection process.

Identifying the selection process is crucial in practice, as it not only explains dependencies in observation but also happens with variations in sample size, leading to unexpected effects. However, the selection bias problem (Heckman, 1978) is overlooked in biology, as it persists beyond the reach of randomized experiments and proves challenging to detect in both experimental and observational studies.

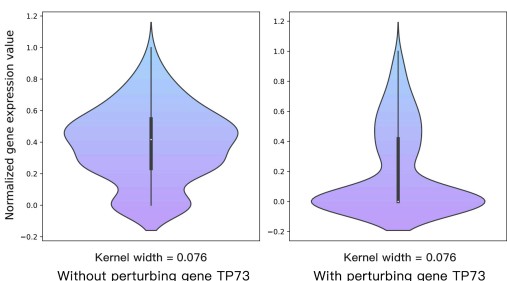

Figure 2: After perturbing gene *TP73*, the distribution of gene *CENPF* change a lot. However, ground truth collected from comprehensive libraries shows they are independent.

Over the past decades, numerous methods have been developed for GRNI, encompassing computational and causal approaches. Computational models, represented by a boolean model, differential equation, gene correlation, and correlation ensemble over pseudo-time, focus on exploring dependencies among genes (Kharchenko et al., 2014; Matsumoto et al., 2017; Li et al., 2021; Deshpande et al., 2022; Li et al., 2024; Nguyen et al., 2021). In contrast, causal models go beyond dependence to uncover the authentic causal relationships within GRNs (Wang et al., 2017; Belyaeva et al., 2021; Zhang et al., 2021). Although some work focused on recovering latent confounders (Xue et al., 2023) and causal relations among genes in GRNI (Chevalley et al., 2022), the selection bias problem has not been considered yet. In causal discovery, exploring the causal process in the presence of selection bias and latent confounders has been challenging. Some fundamental works focused on identifying selection bias under certain parametric assumptions (Kaltenpoth & Vreeken, 2023), studying the identifiability and estimation of functional causal models under the outcome-dependent selection structure condition (Zhang et al., 2016), recovering the conditional probability from selection biased data (Bareinboim et al., 2022). However, these methods are limited to either parametric assumption, i.e., linear Gaussian, or outcome-dependent selection structure, which are unsuitable for the non-parametric setting and the pairwise selection context of GRNI. For causal discovery in the presence of selection bias and latent confounders, the FCI algorithm (Spirtes et al., 1995; Zhang, 2008) aims to discover ancestral relations up to an equivalence class, but significant ambiguities remain for the selection structure. Similarly, some attempts result in ancestral equivalent class limited to graphical properties (Jaber et al., 2019; Rohekar et al., 2021).

In this paper, the problem we focus on is whether it is possible to discover information about the selection process, causal process, and latent confounders from perturbation data. In a traditional view, with a single distribution, it is usually impossible to distinguish dependence induced by the selection process, direct cause, or latent confounders. Surprisingly, by integrating observational data and perturbation data, some interesting findings offer insight into tackling this problem. Specifically, the dependencies arising from causation, selection process, and latent confounders exhibit differences

in symmetry and perturbation effects, making them distinguishable. **Symmetry:** A causal process is asymmetric. Perturbations introduce changes in distribution that only propagate along the causal direction ($X \rightarrow Y$). The selection process on both variables is symmetric, any perturbation on one variable will lead to the distribution change on another ($X \rightarrow S \leftarrow Y$). Latent confounders are also symmetric, however, the distribution change caused by perturbation can not propagate via it ($X \leftarrow L \rightarrow Y$), where $L$ is unobserved. **Perturbation effects:** Moreover, when mixed dependencies, such as cause with the selection process or latent confounders occur, symmetry can no longer be used as the only distinguishing criterion. Interestingly, with additional differences in structures, distinguishable Conditional Independence (CI) patterns between perturbation indicator ($I$) and observed variables emerge as shown in Figure 8 in the Appendix B.

**Contributions.** Based on these properties, our contributions are as follows: **1.** We argue that the long-overlooked selection processes and existing latent confounders explain many confusing dependencies in GRNI. **2.** Usually with a single distribution, it is generally difficult to distinguish selection processes, latent confounders, and causal relations. We should thank the gene perturbation data, which allows for partial recovery of the selection processes, latent confounders, and qualitative structure information from observed dependencies. **3.** Theoretically, with appropriate gene perturbation data, qualitative structure information, selection processes, and latent confounders are partially identifiable without parametric assumptions under mild graphical conditions. **4.** We validate our claims and the effectiveness of our proposed Gene regulatory network Inference in the presence of Selection bias and Latent confounders (GISL) on synthetic and real-world experimental single-cell gene expression data to show its superiority over canonical causal discovery baselines.

## 2 PRELIMINARIES

A Gene regulatory network (GRN) (Levine & Davidson, 2005), focusing on the causal relations and governing gene activities in cell populations, can be represented by a causal model (Ram et al., 2006). The data $X = \{X_1, X_2, ..., X_N\}$ consists of observed variables where each $X_i$ represents an individual gene. Let $\mathcal{G} = (\boldsymbol{V}, \boldsymbol{E})$ be a directed acyclic graph (DAG) model with the vertex set $\boldsymbol{V}$ and edge set $\boldsymbol{E}$, where $\boldsymbol{V} = \{\boldsymbol{X}, \boldsymbol{S}, \boldsymbol{L}, \boldsymbol{I}\}$ encapsulates all observed variables $\boldsymbol{X}$, latent selection variables $\boldsymbol{S}$, latent confounders $\boldsymbol{L}$, and perturbation indicator $\boldsymbol{I}$. Data $D_o$ represents observational data, and $D_{pi}$ is perturbation data with perturbing gene $X_i$.

To introduce the different structures of a causal model, the definition of basic terms should be clear. A causal relation is represented by a directed edge, e.g., $X_i \rightarrow X_j$, where $X_i, X_j \in \boldsymbol{X}$. This is also described as $X_i$ is the parent of $X_j$. In biology, gene $X_i$ regulates gene $X_j$ by intermediate medium, i.e. protein. We also refer to the mechanism underlying a causal relationship as a causal process. If there is a direct path like $X_i \rightarrow \cdots \rightarrow X_j$ between them, $X_i$ is called the ancestor of $X_j$. We denote latent confounder as $L_k \in \boldsymbol{L}$, which is a hidden common cause working on confounded pair in 2.2 contributing to dependence that does not have cause relation. Different from observed variables and latent confounders, the selection process represented by structures ($X_i \rightarrow S_k \leftarrow X_j$) with selection variable $S_k \in \boldsymbol{S}$. We can only observe the data points for which the selection criterion is met, i.e., $S_k = 1$. As $S_k$ is always given, the data distribution actually is $P(\boldsymbol{X}|\boldsymbol{S})$, resulting in spurious dependence between $X_i$ and $X_j$. Some other basic concepts can be found in A.

**Definition 2.1 (Selection bias)** *The distribution $\mathcal{P}$ of the variable in the set $\boldsymbol{V}$ is biased by the selection processes.*

**Definition 2.2 (Confounded pair)** *A pair $(X_i, X_j)$ is a confounded pair, denoted as $(X_i, X_j)_l$. If there exists a latent variable $L_k \in L$ that is the ancestor of a pair $(X_i, X_j)$, and the vertices (apart from $X_i$, $X_j$) on the path between $L_k$ and $X_i$, $X_j$ are latent ($X_i \leftarrow \cdots \leftarrow L_k \rightarrow \cdots \rightarrow X_j$).*

**Definition 2.3 (Selection pair)** *A pair $(X_i, X_j)$ is a selection pair, denoted as $(X_i, X_j)_s$, if it follows the structure $(X_i \rightarrow S_k \leftarrow X_j)$.*

**Definition 2.4 (DAG-inducing path)** *In a DAG G, if a path $p$ between two observed vertices $(X_i, X_j)$ relative $L, S$ is called a DAG-inducing path, if it satisfies the following criteria: 1. There is at least one collider on the path $p$ apart from $(X_i, X_j)$. 2. Every vertex on $p$ is either in $L$ or a*

*collider, and every collider is an ancestor of $X_i$, $X_j$, or a member of S. 3. If the collider is the parent of $S_k \in S$, $X_i$ or $X_j$ is also the parent of $S_k$. Toy examples are shown in 9.*

**Assumption 2.5 (Faithfulness Spirtes et al. (2000))** *Given a DAG $\mathcal{G}$ and distribution $\mathcal{P}$ over the variable set $\mathbf{V}$, $\mathcal{P}$ implies no CI relations not already entailed by the Markov assumption.*

**Assumption 2.6 (Markov)** *Given a DAG $\mathcal{G}$ and distribution $\mathcal{P}$ over the variable set $\mathbf{V}$, every variable $M$ in $\mathbf{V}$ is probabilistically independent of its non-descendants given its parents in $\mathcal{G}$.*

## 3 IDENTIFIABILITY WITHOUT LATENT CONFOUNDERS

Is the structure identifiable when selection coexists with other dependencies as shown in Figure 10? To answer this, we establish the identifiability of the causal structure and partial identifiability of the selection process without any parametric or further structure assumptions.

**Theorem 3.1** *(Partial identifiability) Not all causal structures can be uniquely determined from the available data and assumptions. However, it is possible to determine the set of all possibilities.*

**Theorem 3.2** *(Identifiability) The causal structures are uniquely identified.*

**Theorem 3.3** *(Identifiability and partial identifiability of GISB) Let the observed data consist of a sufficiently large sample generated by the DAG model defined in Section 2. In addition to the faithfulness 2.5 and Markov 2.6 assumptions, suppose there are no latent confounders: $\mathbf{L} = \emptyset$. Then causal processes are identified, selection pairs (selection processes) are partially identified, and selection bias is identified in the causal graph.*

**Motivation and Discussion.** We show the identifiability of the causal process and partial identifiability of the selection process, and develop Algorithm 1 (detailed procedure 3) to achieve it. Usually without extra information, it is difficult to identify the selection process in the non-parametric setting. Both causal and selection processes can generate dependence. FCI can identify certain cases up to the upper bound of information provided by structure properties. Thanks to the perturbation data, the differences between the causal and selection processes emerge, making them distinguishable. With perturbation and observational data, differences in symmetry, perturbation effects, and structure characters among different patterns are reflected in CI patterns between $I$ and observed genes as shown in Figure 3, more details are shown in lines 1, 3 and 5 in Figure 8. Step 2 in the Algorithm 1 deletes condition-independent edges when considering complex cases (multi-path), which significantly improves

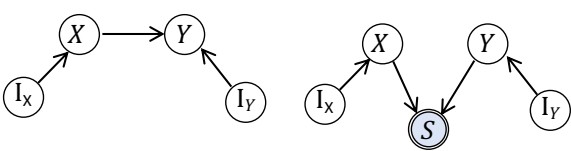

Figure 3: Differences in symmetry and Conditional Independence (CI) patterns: Causation vs. Selection

the efficacy of GISB by reducing the size of condition sets. For example, some complex examples, including both $X \rightarrow N \rightarrow Y$ and $X \rightarrow M \leftarrow Y$, can be identified by conditioning on $N$. Then dependencies can be explained by causal process, selection process, or combinations. Considering the efficacy of the CI test on high-dimensional data, skeleton discovery is not limited to traditional PC. Parallel PC Le et al. (2016), FGES Ramsey et al. (2017), and even computational methods can be applied. With the help of perturbation data, following the rules (the correspondences between CI patterns and structures) shown in Figure 8. Step 3 updates $\mathcal{G}$ and the selection set based on the CI results. Considering multiple paths, Step 4 further corrects the CI patterns with some dependence hidden by the selection process like (b)-3 in Figure 10. By conditioning on $Z$, real structure ($X$ and $Y$ under selection) is identified. Moreover, two cases can distort the CI patterns, e.g., (a)-9 and (b)-8 in Figure 10, resulting in the partial identifiability of the selection process. Where the CI pattern of (a)-9 is distorted due to the Y-structure formed by $X$, $I_Y$, $Y$, and $S$, as $S$ is always given, $X$ and $I_Y$ are dependent, which breaks the CI relations. (b)-8 is because of the DAG-inducing path ($X - Z - Y$), which is always d-connected. Thus, whether $X$ and $Y$ or the descendants are directly under the selection process can not be determined, as they are limited to the true structure of the

---

**Algorithm 1** GISB: Gene Regulatory Network Inference in the Presence of Selection Bias.

---

**Input:** observational data $D_o$, single gene perturbation data $D_p$ for all genes with $D_{pi}$ for gene $X_i$, perturbation indicator $I$.
**Output:** DAG $\mathcal{G} = (\mathcal{V}, \mathcal{E})$, Selection Pairs $\mathcal{S}$.

 1: (*Graph Initialization*) Initialize $\mathcal{G} = (\mathcal{V}, \mathcal{E})$ as a fully undirected graph and list $\mathcal{S}$ as empty.
 2: (*Recovery of regulation skeleton over observational data*) Run skeleton discovery methods on $D_o$.
 3: (*Recovery of the regulation and selection processes over observational and perturbation data*) For each undirected edge of gene pair $(X, Y)$, test the marginal and conditional independence between $I$ of one gene and another gene on augmented data $D_{aug}$ $(D_o + D_{pi})$. Update $\mathcal{G}$ and update $\mathcal{S}$ with identified pairs $(X, Y)_s$.
 4: (*Correct spurious relations*) Repeat Step 3 with conditioning on the subsets of genes on the paths between $X$ and $Y$ in $\mathcal{G}_{aug}$. Update $\mathcal{G}$ and update $\mathcal{S}$ with identified pairs.

---

**DAG-inducing path (selection structures are partially identifiable).** At the same time, they are under selection bias. Fortunately, the causal process is still identified, as the form of Y-structure also needs cause relation, and the DAG-inducing path (undirected edge) does not affect the structure features of the causal process. The comprehensive proof of GISB is in Appendix G.2

## 4 PARTIAL IDENTIFIABILITY WITH LATENT CONFOUNDERS

We previously discussed methods for identifying direct causal relationships and selection mechanisms between genes, assuming no latent confounders. However, in practical scenarios using scRNA-seq data, latent confounders, such as non-gene regulators, transcription factors, and technical covariates, can indeed exist. This raises the question: What can be definitively identified about causal relationships when latent confounders are present?

A most generalized model might include latent variables, perturbation indicators, and observed variables all involved in the selection process (e.g., under some unrecorded experimental conditions, only cells with certain gene expression patterns can successfully receive some gene knockout). Nonetheless, such generalized assumptions often render causal relationships too indeterminate, and thus the results less informative. For example, a direct causal edge $X \to Y$ can generally always be replaced with $X \to S \leftarrow L \to Y$, where $X, Y$ are observed, $L$ is latent, and $S$ is a selection indicator, rendering them indistinguishable in terms of all conditional independence constraints, even with interventional data for $I_X$ and $I_Y$ on both sides.

To address this, we have to adopt a structural assumption: selection processes involve only observed variables, disallowing any causal edges from latent variables ($L$) and perturbation indicators ($I$) to the selection indicators $S$. This assumption is partly justified by the typically lower prevalence of confounders compared to observed variables in scRNA-seq data. Under this framework, what can we identify? We first notice that even without selection and with interventional data, latent confounders can still make the direct causal relations unidentifiable. Consider the case $X \to Z \to Y$ with a latent confounder $L$ pointing to both $Z$ and $Y$ shown in Figure 9 (b). Adding a direct edge $X \to Y$ renders the scenarios equivalent, even if perturbation data $I_X, I_Y$ are available, as the dependence between $X$ and $Y$ cannot solely be explained by $Z$.

This leads us to question whether ancestral causal relationships ($X$ has a direct path to $Y$), instead of direct causal relations, are identifiable with latent confounders. Unfortunately, the answer is still negative. For instance, in the model $S_1 \leftarrow X \leftarrow L \to Y \to S_2$ (Figure 9 (a)), with latent confounder $L$ and selection indicators $S_1, S_2$, whether adding a direct edge $X \to Y$ or not, the two scenarios are unidentifiable, even with interventional data: perturbing $X$ alters $P(Y)$, and this change cannot be solely attributed to $X$; the same happens at the $Y$ side.

Thus, given all the above unidentifiable cases, we conclude that we can only identify the ancestral causal relations and the absence of selection. If all of the following hold: 1) $I_X \not\perp\!\!\!\perp Y$, i.e., the perturbation on $X$ results in a change in $P(Y)$, 2) $I_X \perp\!\!\!\perp Y|X$, i.e., this change is completely explainable by $X$, and 3) $I_Y \perp\!\!\!\perp X$, i.e., the perturbation on $Y$ does not affect $P(X)$, then it can be concluded that $X$ is an ancestor of $Y$, and there is no selection for each of $X, Y$. This is a sufficient condition,

---

**Algorithm 2** GISL: Gene Regulatory Network Inference in the Presence of Selection Bias and Latent Confounders.

---

**Input:** observational data $D_o$, single gene perturbation data $D_p$ for all genes with $D_{pi}$ for gene $X_i$, perturbation indicator $I$.

**Output:** PAG $\mathcal{G} = (\mathcal{V}, \mathcal{E})$, Confounder pairs $\mathcal{L}$, Selection pairs $\mathcal{S}$.

1: (*Graph Initialization*) Initialize $\mathcal{G} = (\mathcal{V}, \mathcal{E})$ as a fully undirected graph and list $\mathcal{L}, \mathcal{S}$ as empty.
2: (*Recovery of regulation skeleton over observational data*) Run skeleton discovery methods on $D_o$.
3: (*Recovery of the regulation, selection processes, and latent confounders from observational and perturbation data*) For each undirected edge of gene pair $(X, Y)$, test the marginal and conditional independence between $I$ of one gene and another gene on augmented data $D_{aug}$ ($D_o + D_{pi}$). Repeat this with conditioning on the subsets of genes on the paths between $X$ and $Y$ in $\mathcal{G}_{aug}$ to remove the spurious dependence and update $\mathcal{G}$. Update $\mathcal{L}, \mathcal{S}$ with identified pairs and mark pairs needed to be corrected.
4: (*Correction*) Further correct those undetermined pairs following the correction rules, and update $\mathcal{G}, \mathcal{L}, \mathcal{S}$.

---

and when the condition is not satisfied, it does not necessarily mean that $X$ is not $Y$'s causal ancestor. Formally, we propose the GISL algorithm and give the following partial identifiability results:

**Theorem 4.1** *Let the observational and perturbation data be sufficient, which are generated by the DAG model defined in Section 2, In addition to the faithfulness2.5 and Markov2.6 assumptions, suppose selection processes can not work on latent variables, i.e., latent variables are not the parent of selection variables. Then the qualitative structure information, selection process, and latent confounders are partially identified in the causal graph.*

**Motivation and Discussion** We show the partial identifiability of the causal process, selection process, and latent confounders and develop Algorithm 2 (detailed procedure 4) to elucidate some interesting laws. When considering the general case, graph structure becomes very complex. Same with Section 3, based on the differences, including symmetry, perturbation effect, and structure characters, reflected in CI patterns between perturbation indicator $I$ and observed genes. The latent structure is shown in Figure 4, and detailed basic patterns in Figure 8 provide insight into distinguishing causal processes, selection structures, and latent confounders from finding unique markers. There are two kinds of cases providing dependence, which blocks us from approaching the true causal structures. One is (a) in Figure 9, as $S$ is given, $I_X$ and $L$ are always dependent, another is the DAG-inducing path (b), (c), and (d) in Figure 9, where dependence occurs as the collider $Z$ works as a chain in other paths between $X, Y$. These result in the spurious causal dependence between $I$ and observed genes, which can not be d-separated A by conditioning operation. Let's start with the algorithm to introduce the interesting laws. Steps 2 and 3 have the same operation as GISB. The correcting rules in Step 4 are as follows: **1.** if the CI pattern changes to another one with less dependence in Figure 8, then change the result to the new one. **2.** If more dependence, usually lines 3 and 5 in Figure8 become a full dependent pattern, which means $I_X$ and $Y$ are dependent and $I_Y$ and $X$ are dependent no matter given $X, Y$ or not, we keep the previous result. As the collider is given, it results in more dependence. With the confusing cases, we found that the result with $X$ cause $Y$ indicates $X$ is $Y$'s ancestor without confounding and selection. The result with confounder pairs indicates $(X, Y)_l$ or $X$ and its ancestor on the path between $X, Y$ form a confounder pair. The result with the selection pair indicates $(X, Y)_s$, or $Y$ form selection pair with the descendants of $X$ on the path between $X, Y$. The selection with cause and confounder with cause indicate the same results as the selection and confounder pair separately. Considering the DAG-inducing path, causal process, selection process, and latent confounders are partially identified limited to true structures and DAG-inducing path. This is because the DAG-inducing path (always d-connected) can pretend to be any structure shown in CI patterns. Selection bias is identifiable. The details of proof can be found in Appendix G.2.

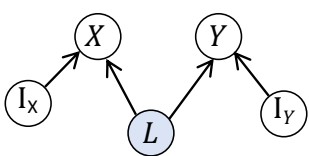

Figure 4: The causal structure of Latent confounders .

## 5 EXPERIMENTS

In this section, we conduct experiments on synthetic and real-world data sets to validate the selection process claim and verify the effectiveness of our proposed GISL in identifying qualitative structures, selection bias, and latent confounders, demonstrating that it is not only theoretically sound but also leads to superior performance in practice.

### 5.1 SYNTHETIC DATASETS

**Parametric setting.** We utilize a simple structure ($X$ cause $Y$ under selection bias) as an illustrative example to elucidate the setting of the parametric model. The synthetic data is generated according to the structure equation model (SEM) as follows:

$$\begin{cases} X = E_x, \\ Y = f(X) + E_y, \\ f_s(X) + f_s(Y) + E_s > 0. \end{cases} \tag{1}$$

where the additive noises, i.e., $E_x$, $E_y$ as well as $E_s$ are assumed to follow Gaussian distribution with randomly selected means and variances. The causal function $f$ and selection function $f_s$ are linear with randomly chosen parameters. Moreover, gene knockout (CRISPR-Case9) and gene knock-up (CRISPRa) technologies working as hard and soft intervention separately are simulated, where hard intervention sets the gene expression value to 0 and soft intervention increases the expression value by adding a uniformly distributed noise. Ground-truth causal structures are generated by Erdös–Rényi model (Erdős et al., 1960) with $d \in \{6, 9, 12, 15, 18\}$ nodes and randomly add 1-3 selection pairs on each causal structure. When considering latent confounders, 1-3 confounder pairs are randomly added. We randomly sample 20 causal structures with 30000 data points for each before selection.

**Non-parametric setting.** Unlike a parametric setting, the non-parametric one considers a complex non-linear causal process. Genes follow the Gaussian distribution with randomly selected means and variances, the causal function and selection function are randomly chosen from linear, square, sin, and tanh functions. Considering the computational efficiency, the ground truth causal structures are generated based on the Erdös–Rényi model with $d \in \{5, 6, 7, 8, 9\}$ nodes and randomly 1-2 selection pairs. 1-2 confounder pairs are randomly added when considering latent confounders. We sample 20 causal structures with 2000 data points before selection for each setting.

**Baselines and evaluation.** To verify the effectiveness of our proposed GISL, we report the structural Hamming distance (SHD), F1 score, precision, and recall to measure the quality of the predictions against ground truth on synthetic data sets compared with canonical baselines. All experiments are from averaging 20 random graphs with CPUs and 12 GB of memory. Without latent confounders, PC (Spirtes & Glymour, 1991), GES (Chickering, 2002), and GIES (Hauser & Bühlmann, 2012) algorithms are set as strong baselines. The GISL outputs a DAG, while the PC, GES, and GIES only find a completed partially directed acyclic graph (CPDAG). To keep consistency at the data level, we use the simple orientation rules (Dor & Tarsi, 1992) implemented by Causal-DAG (Chandler Squires, 2018) to uncover more edges in CPDAG with the help of intervention data. Furthermore, as our algorithm utilizes both observational and perturbation data, while PC and GES only work on observational data, we further utilize perturbation data to assist PC and GES in determining more edges. With latent confounders, the FCI (Spirtes et al., 1995; Zhang, 2008) and ICD (Rohekar et al., 2021) are set as baselines. We report the metrics on PAG compared with baselines.

**Experimental results without latent confounders.** We conduct experiments and a comparative analysis on synthetic data sets to validate our claims about GISB in identifying qualitative structure information, and selection process. First, the priority of introducing perturbation data is evaluated on synthetic data without selection bias as shown in Figure11. Experimental results of GISB and baselines on all evaluation criteria are shown in Figure 5. From Figure 5, we can see that our method shows its superiority over all baselines in different criteria. The reasons are as follows: First, the spurious dependence engendered by the selection bias can not be handled by baselines. Second, even with perturbation data, the causal processes are still not distinguishable under selection bias. This is because the stronger symmetry property of the selection process covers up the asymmetry of the causal process, leading to the unidentifiable existence of qualitative information. However, instead of directly using distribution change, our algorithm models the difference between the asymmetry of

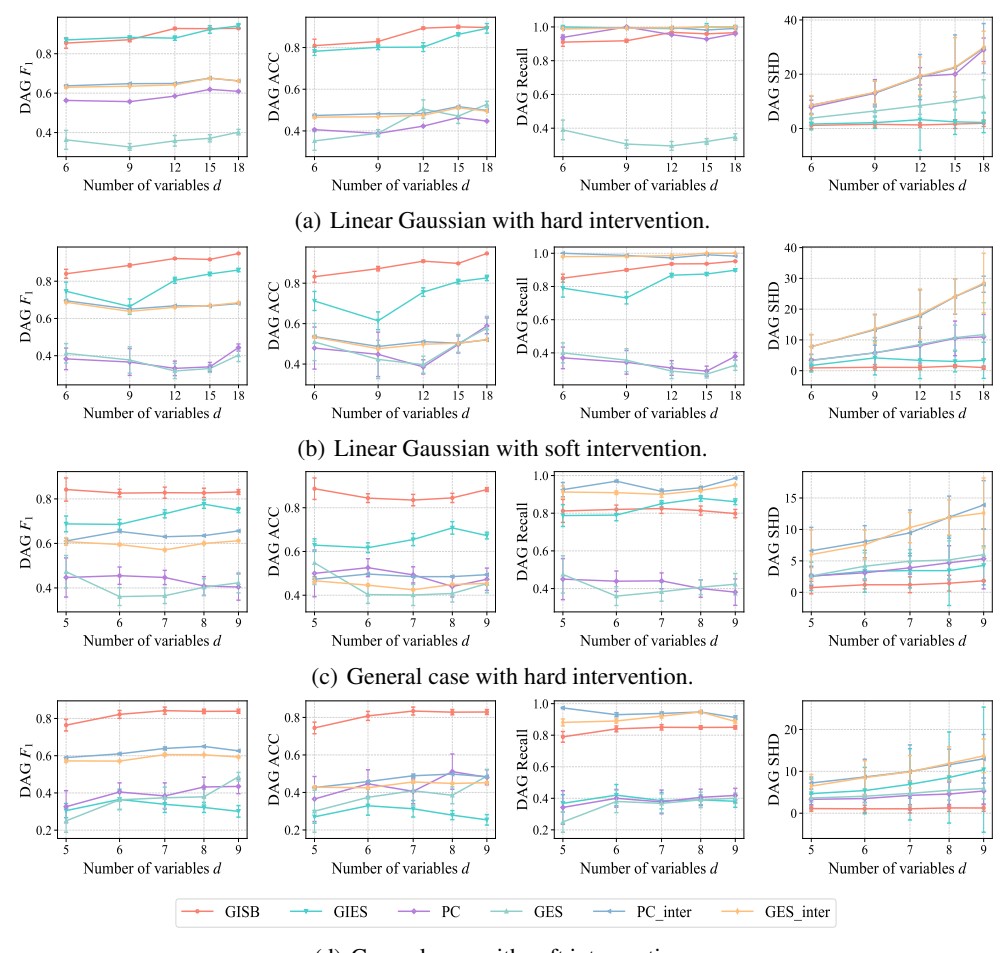

(a) Linear Gaussian with hard intervention.

(b) Linear Gaussian with soft intervention.

(c) General case with hard intervention.

(d) General case with soft intervention.

Figure 5: Experimental results of GISL and strong canonical causal discovery baselines on synthetic data sets, where $PC\_inter$ and $GES\_inter$ indicate that the results are further refined with perturbation data. By rows, we evaluate different variables $d$. By columns, we evaluate DAG $F_1$ ($\uparrow$), DAG ACC ($\uparrow$), DAG Recall ($\uparrow$) and DAG SHD ($\downarrow$).

causation and symmetry of selection by introducing a perturbation indicator $I$ as a surrogate variable. The difference can be expressed in conditional independence relations between the surrogate variable and genes. This design cleverly avoids the drawbacks of baselines and identifies the causal structure for GRNI. Moreover, the presence of selection bias is partially identified. Following the algorithm 1, to start with, we try to distinguish different patterns based on CI test results, but there appear spurious dependencies engendered by selection bias. The reasons are as follows: one is the transitivity of the selection mechanism such as (a)-8 in Figure 10, if the selection process works on the descendant of observed ones, the CI test result shows the existence of selection bias. We tackle it by traversing all subsets of nodes on the paths between $X$ and $Y$. This leads to another case like (a)-6, if the adjacent node forms a V structure with $X$ and $Y$ is given, there will form the illusion of selection bias. Another is the Y structure with the selection variable $S$ as the descendant of the collider, which will break the conditional independent relations by introducing dependence since $S$ is always given.

To evaluate the effectiveness of our proposed GISB in identifying the presence of selection bias, we conduct experiments on causal graphs with d=10 nodes in both linear Gaussian and general cases, considering various numbers of node pairs that are subject to selection processes. We randomly generate 20 causal structures for each setting. Experimental results on all evaluation criteria are shown in Table 2. Overall, with the increasing number of selection processes, GISB still keeps competitive performance even though almost all variables are under selection bias. Due to the partial identifiability of selection bias, the accuracy of identifying selection structures is around 50% to 70%.

Table 1: Experimental results on different numbers of selection processes. *#S* indicates the number of selection process, *SACC* denotes the accuracy of identifying selection structures.

| #S | 1 | 2 | 3 | 4 | 1 | 2 | 3 | 4 |
|---|---|---|---|---|---|---|---|---|
| | Hard intervention | | | | Soft intervention | | | |
| ACC | 88.4±1.1 | 80.5±0.6 | 73.6±0.6 | 65.9±1.4 | 90.4±0.5 | 85.3±0.9 | 80.2±0.7 | 77.5±1.3 |
| Recall | 94.4±0.8 | 93.3±0.6 | 90.5±1.0 | 85.8±1.8 | 93.8±0.5 | 91.1±0.6 | 90.0±0.8 | 88.9±0.5 |
| F1 | 91.2±0.9 | 86.4±0.5 | 81.1±0.7 | 74.4±1.5 | 92.1±0.4 | 88.1±0.7 | 84.6±0.6 | 82.6±0.9 |
| SHD | 1.2±1.1 | 2.1±0.7 | 2.9±1.0 | 4.1±2.6 | 0.9±0.4 | 1.45±0.9 | 2.1±0.9 | 2.4±2.0 |
| SACC | 60.8±17.8 | 65.6±8.2 | 68.4±3.6 | 45.4±6.4 | 70.5±1.6 | 72.1±4.5 | 56.9±6.7.7 | 49.5±14.3 |

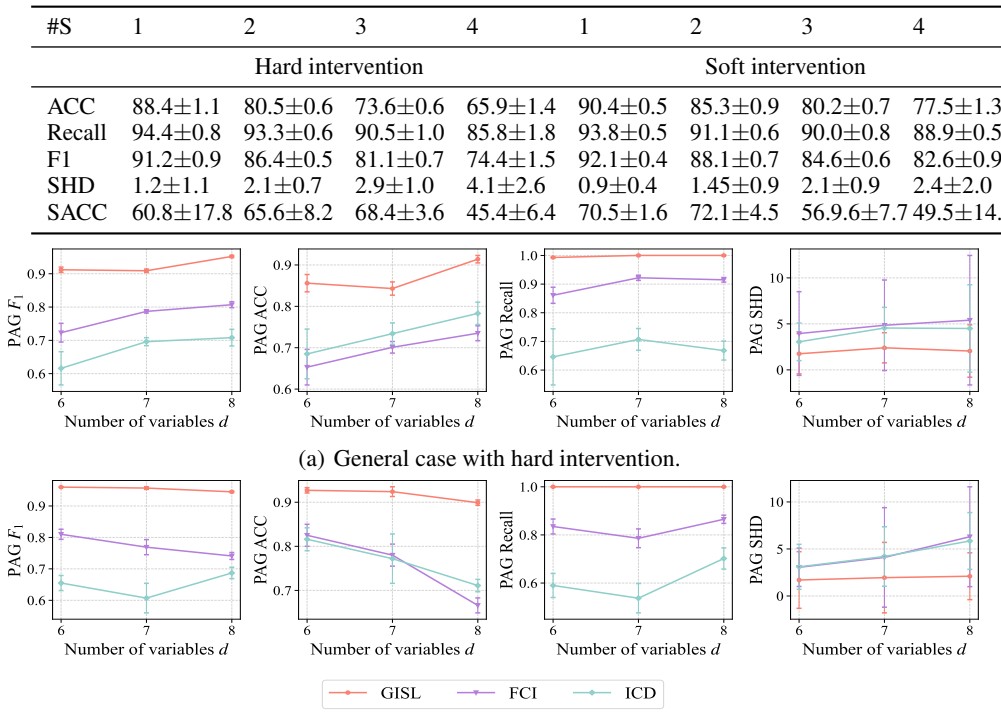

(a) General case with hard intervention.

(b) General case with soft intervention.

Figure 6: Experimental results on PAG $F_1$ (↑), PAG ACC (↑), PAG Recall (↑) and PAG SHD (↓).

**Experimental results with latent confounders.** Experiments are conducted to validate the ability of GISL to identify qualitative structure information, selection processes, and latent confounders. In Figure 6, experimental results on non-parametric settings show the superiorities over FCI and ICD methods. Moreover, the average accuracy of identifying selection structures is $0.708 \pm 0.194$ and $0.910 \pm 0.005$ separately for soft intervention and hard intervention. The average accuracy of identifying latent confounders is $0.841 \pm 0.189$ and $0.654 \pm 0.186$. The reasons are similar to the case that does not consider latent confounders. Integrating differences in symmetry and CI patterns, causal process, selection process, and latent confounders are distinguishable.

## 5.2 REAL-WORLD EXPERIMENTAL DATASETS

**Data availablility** With the advent of next-generation sequencing (NGS) techniques, such as single-cell RNA-sequencing (scRNA-seq), the availability of single-cell data empowers us to conduct more profound analysis of gene expression in biological systems and complex tissues at unprecedented resolution of individual cells (Saliba et al., 2014). Moreover, thanks to the advancement and maturation of gene sequencing and perturbation tools, including CRISPR-Cas9 (Doudna & Charpentier, 2014), CRISPRi (Larson et al., 2013), and CRISPRa Cheng et al. (2013), genes are transformed into viable subjects for causal discovery, providing qualified single-gene observational and perturbation (interventional) data through systematic technique perturb-seq (Adamson et al., 2016; Thomas M. et al., 2019; Dixit et al., 2016).

To examine the efficacy of GISL and validate our claim of the overlooked selection process in a real-world setting, we apply our method to gene expression data collected by Pertrub-seq (Thomas M. et al., 2019). The data are collected from lung carcinoma cells (A-549) with 5045 observable genes and 7353 cells in total. Furthermore, the gene knock-up technique CRISPRa is utilized on cultured cells to enhance the expression value for 105 genes separately, resulting in gene perturbation data. Considering the computational efficiency of CI test methods (general case) and the sparse connect among perturbed genes, we evaluate our

method on a subset of perturbed genes compared with prior knowledge provided by Enrichr (Kuleshov et al., 2016; Chen et al., 2013; Xie et al., 2021) which collects comprehensive libraries. For more detailed information about the real-world setting, please refer to the Appendix H.

In addition, to verify the presence of selection bias, we argue that for each pair of genes, if they are in the presence of selection bias, the number of survived cells varies over perturbing different genes on the premise of culturing the same number of cells. Fortunately, with the CRISPR experimental records organized by DepMap (DepMap, 2023), a cell population dynamics model was proposed for cell proliferation dynamics, where the z-score was designed to show the differences in growth rate between normal cells and perturbed ones. The higher value indicates a significant change in the number of surviving cells following gene perturbation (Dempster et al., 2019; 2021; Pacini et al., 2021). Experimental results of our GISL on a subset of genes with perturbation data as shown in Figure 7. From the figure, one can see that the GISL introduces numerous edges and selection processes that are backed by prior knowledge. For example, Gene pairs ($JUN\ NCL$) and ($JUN\ POU3F2$) are under a selection process with z-scores -0.339, -1.217, 0.252 for $JUN$,$NCL$, and $POU3F2$ respectively. The distribution of the z-score of these genes is shown in Figure 13. Moreover, all edges are collected from Enrichr, black ones are returned by GISL backed by prior knowledge. Besides the efficacy of our method, another superiority of our method is that GISL

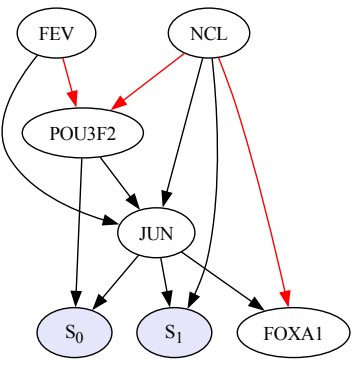

Figure 7: Experimental results on a subset of genes with perturbation data. Red edges are returned by Enrichr (Chen et al., 2013) but not by GISL. Black edges are returned by GISL backed by Enrichr.

is not limited to perturbing all genes. In experimental conditions, we only perturb genes that we want to discover the relationship instead of perturbing genes without guidance, which is time and source-saving.

## 6    CONCLUSION AND DISCUSSION

Rethinking differential gene expression and the observed distributional changes in unregulated genes from gene perturbation data, we argue that the overlooked selection process and the presence of latent confounders significantly bias the performance of gene regulatory network inference (GRNI) in single-cell gene expression data. Many confusing dependent patterns observed from data can be explained by the selection inclusion and latent confounders. Although with a single distribution, it is generally difficult to identify the causal process, selection process, and latent confounders, thanks to gene perturbation data, which provides observations of the differences in symmetry and perturbation effect among them, resulting in distinguishable conditional independent patterns. This motivates us to establish a set of theoretical results demonstrating the partial identifiability of qualitative structure information, latent confounders, and selection processes without any parametric and graphical assumptions. At the same time, we propose a novel GISL algorithm to recover the selection process and latent confounders from causal relations in confusing dependencies among genes. The validity of the presence of the selection process, theoretical claims, and the algorithm's efficacy have been rigorously evaluated on synthetic and real-world data.

**Discussion and Limitations.** In cells, we argue the different intracellular environments, acting as selection mechanisms, constrain the expression of genes. When the environment remains, a selection mechanism is always present. Genes stay in cells with the remaining environment, showing the reasonability of our setting. However, at the algorithmic level, if selection does not occur consistently, whether the intervention happens before or after the selection process will lead to different phenomena. A toy example is designed to introduce this as shown in Figure 14 in Appendix. This interesting discussion is a kind reminder to readers when they apply this algorithm to some specific data, like patients in hospitals. When they recovered, they were still the sample in the dataset. At this time the selection mechanism disappears. Some limitations are listed that are willing to be improved in the future. In our setting, we assume the gene regulatory network is DAG dealing with acyclic relations. The selection process may also work on latent confounders. We focus on the selection process determined by measured genes. Moreover, we focus on the soundness and efficacy of our algorithm and do not pay much attention to the efficiency of the CI test.

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

# Appendix

## A  CONCEPTS

**Definition A.1 (Marginal independence test)** *Check whether two variables $X$ and $Y$ are independent of each other without considering any other variables. Mathematically: $X \perp\!\!\!\perp Y$, meaning $X$ and $Y$ are independent in the overall data distribution.*

**Definition A.2 (Conditional independence test)** *Evaluate whether two variables $X$ and $Y$ are independent given a third variable or set of variables $Z$. Mathematically: $X \perp\!\!\!\perp Y|Z$, meaning $X$ and $Y$ are independent conditioned on $Z$.*

**Definition A.3 (d-separation)** *If every path from a node in $X$ to a node in $Y$ is d-separated by $Z$, then $X$ and $Y$ are always conditionally independent given $Z$.*

## B  EXAMPLE OF DISTINGUISHABLE CI PATTERNS

We list some examples in Figure 8 to show our insight into distinguishing causal process, selection process, and latent confounders given CI patterns. These samples are not complete. Some cases as shown in Figure 9 are unidentifiable in discovering causal processes, as the causal dependencies engendered by the inducing path shown in the second and third cases can not be distinguished from the causal process. Moreover, the first case can be seen as a selection on latent confounders case, where the Y-structure formed by $I_X, X, L, S$ introduces the dependence that can not be d-separated between $I_X$ and $L$, resulting in the spurious CI patterns challenging our algorithm in the identifiability of causal process.

## C  EXAMPLES TO SHOW THE IDENTIFIABILITY OF GISB

Some examples in Figure 10 show insight into identifying different patterns based on CI patterns in the case without latent confounders. Specifically, the causal processes are identifiable, the dotted ones show partial identifiability in the selection process. As the inducing path and Y-structure like (b)-8 and (a)-9, this results in the d-connected path leading to the phenomenon that distribution change can propagate along this path. Then we can not identify the selection structure, at the same time, we can identify the presence of selection bias.

## D  THE PROCEDURE OF ALGORITHM 1

The details of GISB are shown in Algorithm 3. Every step including how to utilize the observational and perturbation data is introduced.

## E  THE PROCEDURE OF ALGORITHM 2

The details of GISL are shown in Algorithm 4. We detail all the steps of the algorithm, similar to how they are listed in GISB.

## F  EXPERIMENTAL RESULTS ON SYNTHETIC DATASET

The experimental results of GISL and baselines on data without selection bias are shown in Figure 11. This shows the superiority of utilizing interventional data to recover causal relations. The distribution change engendered by intervention provides more information in identifying the causal structure.

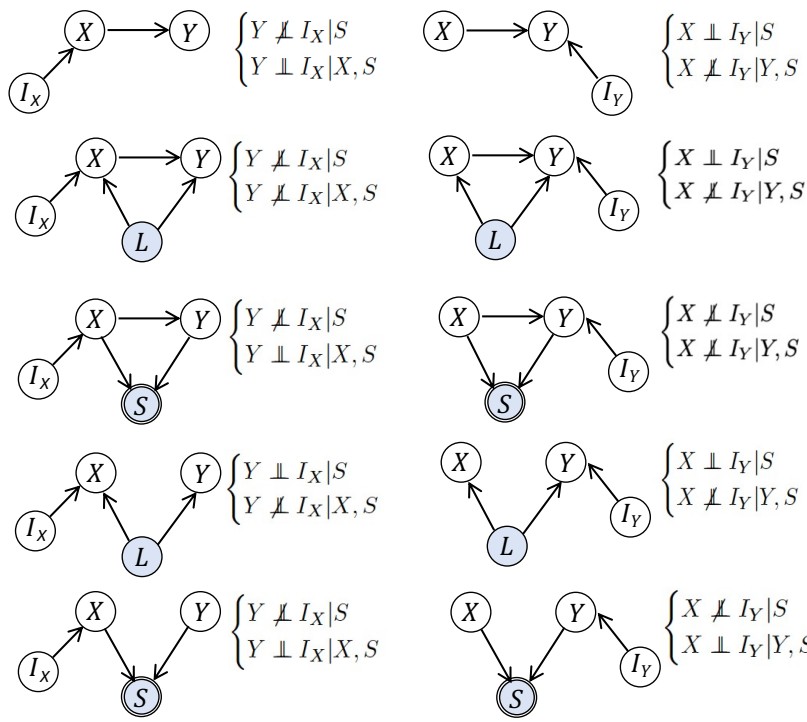

Figure 8: Examples of distinguishable CI patterns, where $S$ is the selection variable indicating the selection process, $L$ is the latent confounder, $X$ and $Y$ are observed variables.

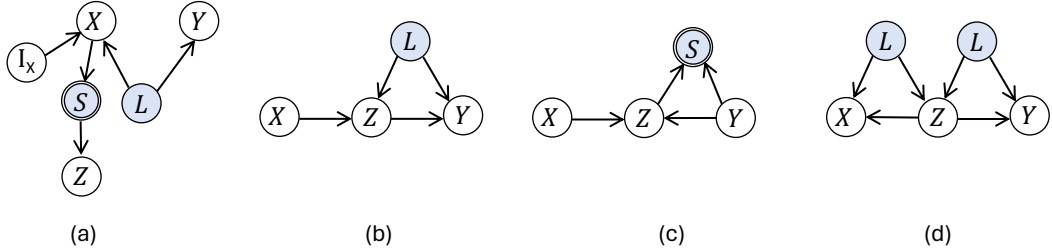

(a)          (b)          (c)          (d)

Figure 9: Non-identifiable cases (DAG-inducing path) correspond to the criteria in 2.4, where the variables $Z$ variables that are colliders in (b), (c), and (d) follow the criteria 2 and 3.

## G  PROOF

### G.1  THEOREM 3.1

**Proof. 1.** The unique CI patterns of causal relation are $X \perp\!\!\!\perp I_Y$ and $Y \not\perp\!\!\!\perp I_X|S$. where $Y \not\perp\!\!\!\perp I_X|S$ needs $X$ and $Y$ are d-connected and no nodes beside $I_X$ point to X. However, $X \perp\!\!\!\perp I_Y$ can only be satisfied when $X - Y - I_Y$ forms a V-structure, which means there is an edge point to $Y$ shown in Figure 12 (a). All in all, between $X$ and $Y$, besides the causal process, if other paths satisfy the previous requirement, there must exist a V-structure, i.e. $X \to Z \leftarrow Y$, and $Z$ is given, as there is an edge point to $Y$, it will form a loop, which conflicts with DAG assumption. However, the V-structure can not point to $Y$ conflicts with the necessary conditions. **2.** Identify the selection process. The selection process needs $X \perp\!\!\!\perp I_Y|Y, S$, and $Y \perp\!\!\!\perp I_X|X, S$ as shown in Figure 12 (b). Any paths between $X, Y$ (point to $X, Y$) apart from the V-structure will conflict with the CI pattern. However, the V-structure is independent given $\emptyset$, which can be distinguished. **3.** Selection with cause. The required structure is shown in Figure 12 (c). $X$ and $I_Y$ are always conditional dependent. It forms a unique Y-structure, i.e., $X \to Y \leftarrow I_Y, Y \to S$. As $S$ is always given, it is mandatory. The proof

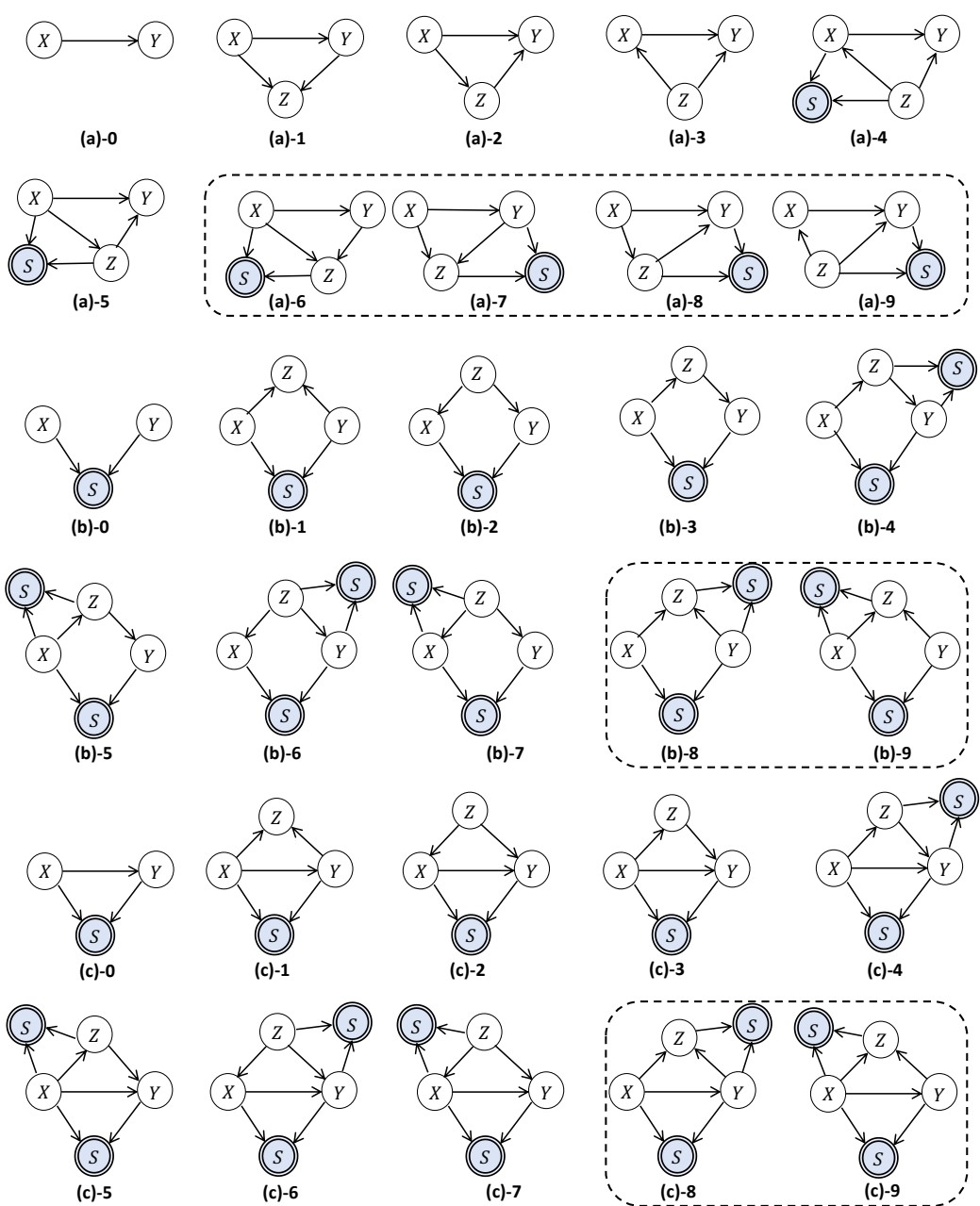

Figure 10: Illustration on all possible cases of causal graphs with three observable variables. The graphs in the dotted box share the same conditional independence relations, and all the other graphs outside the dotted box have different conditional independence relations.

of the causal process is the same as in the previous part. However, the selection process can not be determined between $X, Y$, or the descendant of $X$ and $Y$. Proof done.

## G.2  THEOREM 4.1

**Proof. Causal process:** The conditional independence (CI) pattern of the causal process is illustrated in Figure 8, where it demonstrates that the structure $I_X \to X \to Y$ forms a chain, and $X \to Y \leftarrow I_Y$ represents a collider. If other d-separated paths exist between X and Y, the causal process can still be identified by blocking these paths, which can be achieved by conditioning the vertices on the

---

**Algorithm 3** Concrete procedure of GISB

---

**Input:** observational data $D_o$, single gene perturbation data $D_p$ for perturbed genes with $D_{pi}$ for gene $i$.
**Output:** DAG $\mathcal{G} = (\mathcal{V}, \mathcal{E})$, selection pair $\mathcal{S}$.
Initialize $\mathcal{G} = (\mathcal{V}, \mathcal{E})$ as fully-connected graph. List $\mathcal{S}$ selection pairs as empty.
**All $s \in S$ is given.**
**for** any pair of genes $(x, y)$ **in** $\mathcal{V}$ **do**
  **if** $x \perp\!\!\!\perp y|$ any subset of $\mathcal{V}$-$\{x, y\}$ on $D_o$ **then**
    remove the edge between $x$ and $y$ from $\mathcal{E}$, update $\mathcal{G}$.
  **end if**
**end for**
Introduce surrogate variable (perturbation indicator) $I = 0$ for $D_o$ and $I = 1$ for $D_p$.
**for** edge between genes $(x, y)$ **in** $\mathcal{E}$ **do**
  Construct $D_x$ by concatenating $D_o$ with $I_X = 0$ and $D_{px}$ with $I_X = 1$. Similarly, construct $D_y$.
  **if** $x \perp\!\!\!\perp I_Y|s$ on $D_y$ **then**
    $x$ cause $y$, update $\mathcal{G}$.
  **else if** $y \perp\!\!\!\perp C_x|s$ on $D_x$ **then**
    $y$ cause $x$, update $\mathcal{G}$.
  **else if** $x \not\perp\!\!\!\perp C_y|s$; $x \perp\!\!\!\perp C_y|y, s$ on $D_y$ and $y \not\perp\!\!\!\perp C_x|s$; $y \perp\!\!\!\perp C_x|x, s$ on $D_x$ **then**
    $x$ and $y$ under selection without cause, update $\mathcal{S}$ with $(x, y)$.
  **else**
    **for** subsets $t$ of nodes on the paths form $x$ to $y$ **do**
      **if** $x \perp\!\!\!\perp C_y|t, s$ on $D_y$ **then**
        $x$ cause $y$, update $\mathcal{G}$.
      **else if** $y \perp\!\!\!\perp C_x|t, s$ on $D_x$ **then**
        $y$ cause $x$, update $\mathcal{G}$.
      **else if** $x \not\perp\!\!\!\perp C_y|t, s$; $x \perp\!\!\!\perp C_y|t, y, s$ on $D_y$ and $y \not\perp\!\!\!\perp C_x|t, s$; $y \perp\!\!\!\perp C_x|t, x, s$ on $D_x$ **then**
        $x$ and $y$ are under selection without cause, update $\mathcal{S}$ with $(x, y)$.
      **else if** $x \not\perp\!\!\!\perp C_y|t, s$ and $x \not\perp\!\!\!\perp C_y|t, y, s$ on $D_y$ **then**
        $x$ cause $y$ under selection, update $\mathcal{G}$, update $\mathcal{S}$ with $(x, y)$.
      **else if** $y \not\perp\!\!\!\perp C_x|t, s$ and $y \not\perp\!\!\!\perp C_x|t, x, s$ on $D_x$ **then**
        $y$ cause $x$ under selection, update $\mathcal{G}$, update $\mathcal{S}$ with $(x, y)$.
      **end if**
    **end for**
  **end if**
**end for**
**return** DAG $\mathcal{G} = (\mathcal{V}, \mathcal{E})$, selection pairs $\mathcal{S}$.

---

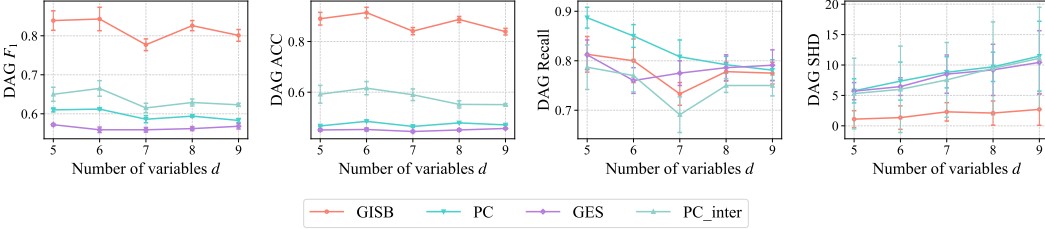

Figure 11: Experimental results of GISB and baselines on synthetic dataset without selection bias.

paths. However, cases involving DAG-inducing paths, such as those shown in Figure 9 (b), result in d-connected paths between X and Y, which is the same as the causal process in CI patterns but is different in structures. Moreover, structures shown in Figure 9 (a) break the collider $I_X \rightarrow X \leftarrow L \rightarrow Y$, working like a causal process as well, leading to partial identification of the causal process.

**Latent confounders:** The unique structure involving latent confounders is represented by the collider configuration $I_X \rightarrow X \leftarrow L \rightarrow Y \leftarrow I_Y$. If there are d-separated paths between X and Y, the latent

---

**Algorithm 4** Concrete procedure of GISL

---

**Input:** observational data $D_o$, single gene perturbation data $D_p$ for perturbed genes with $D_{pi}$ for gene $i$.
**Output:** PAG $\mathcal{G} = (\mathcal{V}, \mathcal{E})$, latent pairs $\mathcal{L}$, selection pairs $\mathcal{S}$.
Initialize $\mathcal{G} = (\mathcal{V}, \mathcal{E})$ as fully-connected graph.
correct-set = []
condition-set = []
**for** any pair of genes $(x, y)$ **in** $\mathcal{V}$ **do**
    **if** $x \perp\!\!\!\perp y|$ any subset of $\mathcal{V} - \{x, y\}$ on $D_o$ **then**
        remove the edge between $x$ and $y$ from $\mathcal{E}$, update $\mathcal{G}$.
    **end if**
**end for**
Introduce surrogate variable (perturbation indicator) $I = 0$ for $D_o$ and $I = 1$ for $D_p$.
**for** edge between genes $(x, y)$ **in** $\mathcal{E}$ **do**
    Construct $D_x$ by concatenating $D_o$ with $I_X = 0$ and $D_{px}$ with $I_X = 1$. Similarly, construct $D_y$.
    **if** $x \perp\!\!\!\perp I_Y|s; x \not\perp\!\!\!\perp I_Y|y, s;$ on $D_y, y \not\perp\!\!\!\perp I_X|s; y \perp\!\!\!\perp I_X|x, s$ on $D_x$ **then**
        $x$ cause $y$, update $\mathcal{G}$.
    **else if** $x \not\perp\!\!\!\perp I_Y|s; x \perp\!\!\!\perp I_Y|y, s;$ on $D_y, y \perp\!\!\!\perp I_X|s; y \not\perp\!\!\!\perp I_X|x, s$ on $D_x$ **then**
        $y$ cause $x$, update $\mathcal{G}$.
    **else if** $x \not\perp\!\!\!\perp I_Y|s; x \perp\!\!\!\perp I_Y|y, s;$ on $D_y, y \not\perp\!\!\!\perp I_X|s; y \perp\!\!\!\perp I_X|x, s$ on $D_x$ **then**
        $x$ and $y$ under selection without cause, update $\mathcal{S}$ with $(x, y)$.
    **else if** $x \not\perp\!\!\!\perp I_Y|s; x \perp\!\!\!\perp I_Y|y, s;$ on $D_y, y \not\perp\!\!\!\perp I_X|s; y \not\perp\!\!\!\perp I_X|x, s$ on $D_x$ **then**
        $x$ cause $y$ under selection bias, update $\mathcal{S}$ with $(x, y)$, correct-set add $(x, y)$, condition-set add ('S-C').
    **else if** $x \not\perp\!\!\!\perp I_Y|s; x \perp\!\!\!\perp I_Y|y, s;$ on $D_y, y \not\perp\!\!\!\perp I_X|s; y \not\perp\!\!\!\perp I_X|x, s$ on $D_x$ **then**
        $y$ cause $x$ under selection bias, update $\mathcal{S}$ with $(x, y)$, correct-set add $(y, x)$, condition-set add ('S-C').
    **else if** $x \perp\!\!\!\perp I_Y|s; x \not\perp\!\!\!\perp I_Y|y, s;$ on $D_y, y \not\perp\!\!\!\perp I_X|s; y \not\perp\!\!\!\perp I_X|x, s$ on $D_x$ **then**
        $x$ cause $y$ under latent confoudner, update $\mathcal{L}$ with $(x, y)$, correct-set add $(x, y)$, condition-set add ('S-L').
    **else if** $x \not\perp\!\!\!\perp I_Y|s; x \not\perp\!\!\!\perp I_Y|y, s;$ on $D_y, y \perp\!\!\!\perp I_X|s; y \not\perp\!\!\!\perp I_X|x, s$ on $D_x$ **then**
        $y$ cause $x$ under latent confoudner, update $\mathcal{L}$ with $(x, y)$, correct-set add $(y, x)$, condition-set add ('S-L').
    **else if** $x \perp\!\!\!\perp I_Y|s; x \not\perp\!\!\!\perp I_Y|y, s;$ on $D_y, y \perp\!\!\!\perp I_X|s; y \not\perp\!\!\!\perp I_X|x, s$ on $D_x$ **then**
        $x$ and $y$ under latent confounder without cause, update $\mathcal{L}$ with $(x, y)$.
    **else**
        correct-set add $(y, x)$, condition-set add ('C-D'). correct-set add $(x, y)$, condition-set add ('C-D').
    **end if**
**end for**

---

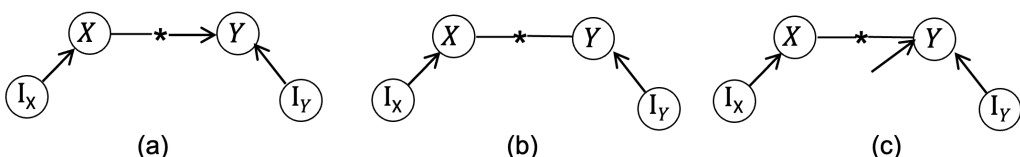

(a)          (b)          (c)

Figure 12: Required structure for causal relation, latent confounders, and selection process, Where * means the always d-connected node.

confounders can be identified, as the CI pattern remains unaffected when these d-separated paths are blocked. However, cases with DAG-inducing paths, such as the scenario depicted in Figure 9 (d), cannot be identified. This is because the d-connected paths between X and Y mimic the same unique structures associated with latent confounders. Nonetheless, latent confounders must exist within the d-connected paths, leading to partial identifiability of these confounders.

**for** index pair in enumerate correct-set **do**
    $x, y$ = pair[0], pair[1]
    **for** all subsets $s_a$ of nodes on the paths from $x$ to $y$ on the path from $x$ to $y$ **do**
        Given subset $s_a$
        **if** condition-set[index] is 'S-C' **then**
            **if** $x \not\perp\!\!\!\perp I_Y|s$; $x \not\perp\!\!\!\perp I_Y|y, s$; on $D_y$, $y \not\perp\!\!\!\perp I_X|s$; $y \not\perp\!\!\!\perp I_X|x, s$ on $D_x$ **then**
                continue
            **else if** $x \not\perp\!\!\!\perp I_Y|s$; $x \not\perp\!\!\!\perp I_Y|y, s$; on $D_y$, $y \not\perp\!\!\!\perp I_X|s$; $y \perp\!\!\!\perp I_X|x, s$ on $D_x$ **then**
                continue
            **else if** $x \perp\!\!\!\perp I_Y|s$; $x \not\perp\!\!\!\perp I_Y|y, s$; on $D_y$, $y \not\perp\!\!\!\perp I_X|s$; $y \perp\!\!\!\perp I_X|x, s$ on $D_x$ **then**
                $x$ cause $y$, update $\mathcal{G}$, continue
            **else**
                remove edge between $x$ and $y$, update $\mathcal{G}$
                break
            **end if**
        **else if** condition-set[index] is 'S-L' **then**
            **if** $x \not\perp\!\!\!\perp I_Y|s$; $x \not\perp\!\!\!\perp I_Y|y, s$; on $D_y$, $y \not\perp\!\!\!\perp I_X|s$; $y \not\perp\!\!\!\perp I_X|x, s$ on $D_x$ **then**
                continue
            **else if** $x \perp\!\!\!\perp I_Y|s$; $x \not\perp\!\!\!\perp I_Y|y, s$; on $D_y$, $y \not\perp\!\!\!\perp I_X|s$; $y \not\perp\!\!\!\perp I_X|x, s$ on $D_x$ **then**
                continue
            **else if** $x \perp\!\!\!\perp I_Y|s$; $x \not\perp\!\!\!\perp I_Y|y, s$; on $D_y$, $y \not\perp\!\!\!\perp I_X|s$; $y \perp\!\!\!\perp I_X|x, s$ on $D_x$ **then**
                $x$ cause $y$, update $\mathcal{G}$, continue
            **else**
                remove edge between $x$ and $y$, update $\mathcal{G}$
                break
            **end if**
        **else if** condition-set[index] is 'C-D' **then**
            **if** $x \perp\!\!\!\perp I_Y|s$; $x \not\perp\!\!\!\perp I_Y|y, s$; on $D_y$, $y \not\perp\!\!\!\perp I_X|s$; $y \perp\!\!\!\perp I_X|x, s$ on $D_x$ **then**
                $x$ cause $y$, update $\mathcal{G}$.
            **else if** $x \not\perp\!\!\!\perp I_Y|s$; $x \perp\!\!\!\perp I_Y|y, s$; on $D_y$, $y \not\perp\!\!\!\perp I_X|s$; $y \perp\!\!\!\perp I_X|x, s$ on $D_x$ **then**
                $x$ and $y$ under selection without cause, update $\mathcal{S}$ with $(x, y)$.
            **else if** $x \not\perp\!\!\!\perp I_Y|s$; $x \not\perp\!\!\!\perp I_Y|y, s$; on $D_y$, $y \not\perp\!\!\!\perp I_X|s$; $y \perp\!\!\!\perp I_X|x, s$ on $D_x$ **then**
                $x$ cause $y$ under selection bias, update $\mathcal{S}$ with $(x, y)$,.
            **else if** $x \perp\!\!\!\perp I_Y|s$; $x \not\perp\!\!\!\perp I_Y|y, s$; on $D_y$, $y \not\perp\!\!\!\perp I_X|s$; $y \not\perp\!\!\!\perp I_X|x, s$ on $D_x$ **then**
                $x$ cause $y$ under latent confoudner, update $\mathcal{L}$ with $(x, y)$,.
            **else if** $x \perp\!\!\!\perp I_Y|s$; $x \not\perp\!\!\!\perp I_Y|y, s$; on $D_y$, $y \perp\!\!\!\perp I_X|s$; $y \not\perp\!\!\!\perp I_X|x, s$ on $D_x$ **then**
                $x$ and $y$ under latent confounder without cause, update $\mathcal{L}$ with $(x, y)$,.
            **end if**
        **end if**
    **end for**
**end for**
**return** PAG $\mathcal{G} = (\mathcal{V}, \mathcal{E})$, selection pairs $\mathcal{S}$, latent confounders $\mathcal{L}$.

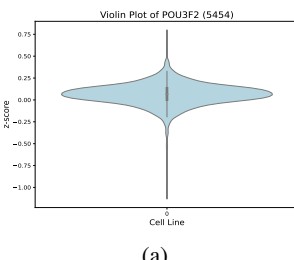 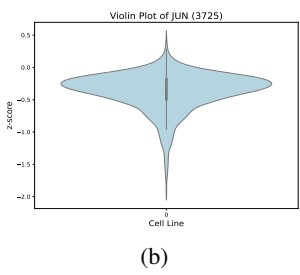 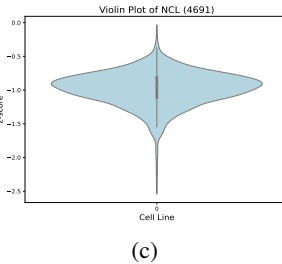

| (a) | (b) | (c) |

Figure 13: An example of the distribution of z-scores of genes among all cell lines.

Table 2: Experimental results of GISB and computational baselines on synthetic data

| Methods | Acc | Recall | F1 | SHD |
|---|---|---|---|---|
| GISB | 94.7±0.01 | 95.1±0.01 | 94.9±0.01 | 1.0±0.54 |
| PIDC Chan et al. (2017) | 5.5±0 | 1.0±0 | 10.5±0 | 153±0 |
| PPCOR Kim (2015) | 5.5±0 | 1.0±0 | 10.5±0 | 153±0 |

**Selection process:** The unique structure of the selection process, characterized by the paths $I_X \to X \to S$ and $I_Y \to Y \to S$, leads to distinguishable CI patterns, as illustrated in Figure 8. Similarly, cases involving d-separated paths can be identified. However, in scenarios with DAG-inducing paths, such as the one shown in Figure 9 (c), the d-connected paths between X and Y exhibit the same structures, i.e., $I_X \to X \to$ and $I_Y \to Y \to$. Furthermore, the d-connected property in these cases is identical to that of the selection process, leading to the partial identifiability of the selection process. Consequently, the selection process is only partially identified.

## H  EXPERIMENTAL SETTING OF REAL-WORLD DATASET

In the real-world dataset, not all the perturbed genes are reported in the Enrichr, as some genes can not be perturbed or processed by biological tools like ChIP-Seq. This leads to the sparse connection among perturbed genes. To illustrate the regulatory relationships in a graphical format, we randomly select a subset of genes that effectively highlight the key interactions. Then GISL is applied to recover qualitative structure information and selection processes. For evaluating the selection process, a z-score is utilized to verify the existence of the selection process. Z-score represents the ratio of the growth ratio between perturbed genes and normal ones. The changes in growth rate indicate the variation in sample size, which is aligned with the property of the selection process. Then, it can be used as an evaluation tool. Some distributions of z-score of the genes we reported are shown in Figure 13. From the figure, we can see that these genes exhibit differences in growth rates between the perturbed one and the normal one, which means under the selection process. In some cell lines, it does not change, which gene is not under selection in this cell. This is consistent with the reason why we explained about the differential gene expression.

## I  COMPARED WITH COMPUTATIONAL METHODS

We rethink the gene regulatory network inference from a causal view and focus on identifying the causal process, latent confounders, and selection process. The setting and the output are different from computational methods, which can not handle the dependence engendered by latent confounders and selection bias. Experimental results of GISB and computational methods on synthetic data are good examples to illustrate this as shown in 2. From the table, we can see that with selection bias, computational methods fail to identify causal relations. This is because the selection process influences not only the variables it directly targets but also those connected along the same path.

## J  DISCUSSION

From Figure 14, we can see that in the left figure, $X_1 \perp\!\!\!\perp X_2$. When intervention is done after selection, and selection does not work anymore, this results in the scatter plot of the middle one. The distribution of $\mathbb{P}(Y|X)$ changes. The last one shows that selection remains. It looks like $\mathbb{P}(Y|X)$ changes from the scatter plot. However, the CI test pattern keeps, i.e., $Y \not\perp\!\!\!\perp I_X|S$ and $Y \perp\!\!\!\perp I_X|X, S$, this is because the increased value range of $X$ is only related to intervention operation ($I_X = 1$)

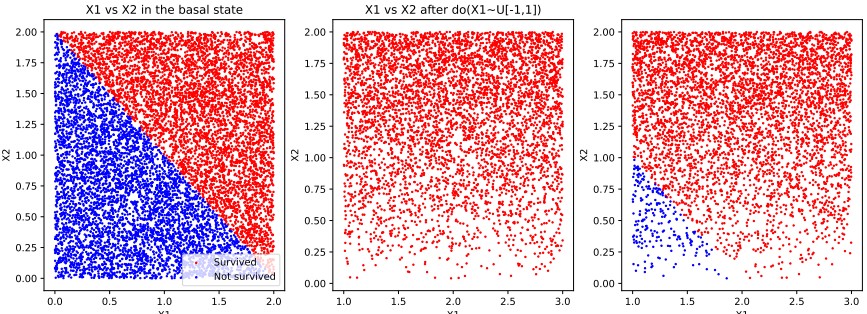

Figure 14: Consistent selection vs. one-time selection.

