# OpenReview forum: "Gene Regulatory Network Inference in the Presence of Selection Bias and Latent Confounders"
_ICLR.cc/2025/Conference — Submitted to ICLR 2025_

### Official Review · Reviewer_qede · 2024-11-01

**Soundness:** 2
**Presentation:** 1
**Contribution:** 1
**Rating:** 3
**Confidence:** 4

**Summary:**

This paper studies the task of inferring gene regulatory networks from perturbation data. It focuses on the issue of latent confounding and selection bias. Specifically, as illustrated in Figure 1, the work attempts to argue that many causal dependencies in gene regulatory network may be spurious due to a selection bias, where the expression of one gene depends on the expression of another gene, and this may be discovered through symmetric effects under perturbations. The paper proposes an algorithm based on testing conditional-independencies for discovering the causal graph and “selection pairs”, which uses the PC algorithm to learn a DAG skeleton from observational data and then post-processes the skeleton to identify selection pairs via the perturbation data. An extension under latent confounding is also proposed. The work provides theoretical identifiability results for both settings that are surprisingly strong, showing that all “causal processes” are identifiable without parametric assumptions.

**Strengths:**

- The basic idea, which is illustrated in the toy examples (e.g. Fig 1), is very sensible. If a measurement bias exists, it can induce “spurious” causal dependencies that even hold under perturbations. I am not aware of prior work in the causal inference literature that discusses this
- The theoretical identifiability results (while not stated very formally, see below) are a strength of the work

**Weaknesses:**

-	The work does not adequately discuss prior work. There is no section in the paper that covers how this work fits into existing works, both in the causal inference literature and computational biology. It may be that no prior work studied this particular artefact of selection bias, but there are probably many related works that are worth discussing and putting into context.
-	The work provides no convincing evidence or justification of the fact that selection bias is an issue in single cell transcriptomics. Which biological or measurement artefact could explain a selection bias of the form in Figure 1 in real-world experimental data? Simply because this synthetic example can induce symmetric perturbation effects does not imply that real-world gene regulatory networks really suffer from this issue. The reasoning in the experiments section (Section 5.2, line 490-499) is shaky. Hinting at differences in cell survival for different perturbations, based on prior data of other experimental studies, is not strong evidence that “selection bias” exists. The work uses z-scores from the DepMap study to show differences in growth rates of cells under different perturbations. However, this could simply be an artefact of the wet-lab experiment or the (causal) regulatory changes induced by a specific perturbation. Overall, while the toy idea makes sense, I don’t find it convincing that the toy idea selection bias is happening in real systems, and I think it requires a lot more justification.
-	Motivating example (Figure 1): couldn’t this "artefact" or selection bias also be viewed as a causal mechanism? Given the selection threshold, the value of X affects the distribution of Y. While we may view this as selection bias, I don’t see why this could not also be seen as a “causal “dependency. In my eyes, if the distribution of Y changes if we perturb X, this is what one may reasonably define as what causation means. How else would we define causality?
-	One of the key challenges in inferring gene regulatory networks from data is the high dimensionality. Like the vanilla PC algorithm, it seems unrealistic to expect a method based on conditional independence testing to scale to specifically the GRN inference problem in practice. Even though the PC algorithm is classical at this point, it is not widely used in computational biology. This is reflected in the experiments (Section 5), where all experiments are performed on less than 20 variables. Since this work focuses particularly on the application in biology, this is not a convincing algorithm for practical applications.
-	The Theorem statements are not rigorous. Theorems 3.1 and 4.1. are very vague descriptions. What is “partially identifiable”? (And what is not?) What are “causal processes”? What does it mean for the data to be “sufficient” or the sample to be “sufficiently large”? How much perturbational data is needed? From the theorems, it is not clear what is proven formally in terms of the formalisms in Section 2. Also, how do these results fit into the context of existing known identifiability results? Can it be seen as a general or special case of other results?
-	The Appendix contains a lot of grammatical language errors and is thus hard to comprehend (e.g. Section F)

**Questions:**

-	Following the line of reasoning in the paper (e.g. following “Symmetry” in line 107), the work argues that causal dependencies are asymmetric, but selection biases are symmetric. I don’t agree with this claim because it assumes that gene regulatory networks are DAGs, which is known to be not hold in general in practice. If perturbing X affects Y and vice versa, this could be due to a feedback cycle between X and Y, and does not necessarily have to occur because of a selection bias.
-	What are “confusing dependencies” in GRNI? (line 117)? Simply because they are not in line with alternative experimental databases (like Enrichr) does not imply that a dependency is spurious. Lots of different circumstances could explain this.
-	What is the data-generating process underlying the causal model described in Section 2? Is it a Bayesian network? The graph alone is not sufficient to describe a distribution over X.
-	What is a “selection bias” formally? Is it that we discard samples if a criterion (an inequality constraint?) is not met, as in the toy example of the introduction? Section 2 views it more as an additional node in the graph. Overall, the notion of selection “bias” is not rigorously described, which makes it hard to understand the theoretical results.

---

> ### Author Response · Authors · 2024-11-25
> **Responses to Reviewer qede**
>
> We thank your time, valuable feedback, and recommendations for improvement.
>
> **Q1:** The work does not adequately discuss prior work.
>
> **A1:** We discuss Gene Regulatory Network Inference in the view of causal discovery with latent confounders and selection bias, which is a super challenging problem in the community of causal discovery. The survey paper on computational methods and some tries in causal discovery is shown in Lines 81-100. As we know, besides FCI, this is the first work that can do causal discovery in the general setting while considering latent confounders and selection bias. If you know other methods can handle this, please let us know. Thanks.
>
> **Q2:** The work provides no convincing evidence or justification of the fact that selection bias is an issue in single-cell transcriptomics.
>
> **A2:** The phenomenon in real-world data shown in Figure 2 and the corresponding description in Lines 50-71 introduced why it inspired us to consider the selection process. Please check Thanks!
>
> The z-scores from DepMap show differences between perturbed cells and normal ones in growth rate, which is treated as evidence to further verify the existence of the selection process. The phenomenon aligns with the property of the selection process in changing of sample size. More analysis of the reliability of the z-score can be found in Appendix H. The z-score is calculated for all cell lines, allowing us to rule out the possibility that it simply is an artifact of the wet-lab experiment
>
> **Q3:** Motivating example (Figure 1): couldn’t this "artifact" or selection bias also be viewed as a causal mechanism?
>
> **A3:** Good question! The selection process can not be viewed as a causal mechanism, although they look similar in distribution change. However, the differences are as follows:
> 1. In structure, causation is asymmetric, and selection is symmetric.
> 2. In biology, genes can be overexpressed via causal function. The selection process can not. For example, the expression value range of Gene A  and B is [5, 15], [5, 15] separately. If there is no causal relation between A and B, the selection process is 10<A+B<20, no matter how to perturb A or B, the value range of A and B can only shrink.
> 3. Changing in sample size only happens in the selection process.
>
> **Q4:** Efficacy of the algorithm on data with high dimensions.
>
> **A4:** Our proposed algorithms have two stages, i.e., skeleton discovery and pair-wise causal discovery. The concern of efficacy on high-dimensional data happens in the skeleton discovery stage. The algorithm is not limited to PC for skeleton discovery. Parallel methods like parallel PC and FGES can eliminate the concern. The running time of FGES on synthetic (5806 genes) and real-world (5045 genes) datasets is 195 minutes and 335 minutes respectively, which prove the feasibility of GISL on high-dimensional data.
>
> **Q5:** What is “partially identifiable”? (And what is not?) What are “causal processes”?
>
> **A5:**  We add more description in Section 2 and Section 3 to introduce these terms. In Theorems 3.1 and 4.1, partially identifiable means in the general case (loose assumption), the detected structures based on CI patterns are not unique but limited to true structure or DAG-inducing path. The causal process is the causal mechanism that connects a cause to an effect.
>
> **Q6:** From the theorems, it is not clear what is proven formally in terms of the formalisms in Section 2. Also, how do these results fit into the context of existing known identifiability results? Can it be seen as a general or special case of other results?
>
> **A6:** We prove the identifiability and partial identifiability of GISB and GISL. With the terms in Section 2, the structure (patterns between variables) can be described formally, which is helpful for readers to understand in proof. Some terms and proof ideas like proof by contradiction are inspired by [1]. This is a new setting and modeling to combine observational and perturbation data by inviting the perturbation indicator. This theorem is more general compared with other results.
>
> [1] Zhang, Jiji. "On the completeness of orientation rules for causal discovery in the presence of latent confounders and selection bias." _Artificial Intelligence_ 172.16-17 (2008): 1873-1896.
>
> **Q7:** Grammatical language errors in Appendix
>
> **A7:** Thanks for pointing it out. We carefully check the paper and fix all errors.

---

> > ### Author Response · Authors · 2024-11-25
> >
> > **Q8:** The DAG assumption for GRNI.
> >
> > **A8:** The DAG assumption is commonly used in GRNI, which is particularly valuable when dealing with noisy and high-dimensional gene expression data [2, 3].
> >
> > **Theoretically**, when involving cycles, the equivalence of the fundamental global and local Markov conditions characteristics of DAGs no longer holds.  The complete procedure might be rather complex, as shown in [4]. Given that the development in the **cyclic case is technically highly nontrivial** (e.g., the equivalence class is just characterized very recently with observational data [5]), we go further and focus on the acyclic case in this paper with both observational and interventional data, and hope to extend it to the cyclic case in the future, after the acyclic case is crystal clear.
> >
> > [2] Bernaola, Niko, et al. "Learning massive interpretable gene regulatory networks of the human brain by merging Bayesian networks." _PLOS Computational Biology_ 19.12 (2023): e1011443.
> >
> > [3] Belyaeva, Anastasiya, Chandler Squires, and Caroline Uhler. "DCI: learning causal differences between gene regulatory networks." _Bioinformatics_ 37.18 (2021): 3067-3069.
> >
> > [4] Richardson, Thomas S. "Discovering cyclic causal structure." _Uncertainty in Artificial Intelligence_. PMLR, 1996.
> >
> > [5] Claassen, Tom, and Joris M. Mooij. "Establishing Markov equivalence in cyclic directed graphs." _Uncertainty in Artificial Intelligence_. PMLR, 2023.
> >
> > **Q9:**  What are “confusing dependencies” in GRNI? (line 117)?
> >
> > **A9:**  In a causal view, dependencies that are not generated by the causal process are spurious, as we only care about causal (regulatory) relations. Dependencies come from causal processes, latent confounders, and selection bias. Could you introduce more about "Lost of different circumstances could explain this". In our setting, we do not think there are other structures besides latent confounders and selection bias can result in spurious dependencies from a causal view.
> >
> > **Q10:** What is the data-generating process underlying the causal model described in Section 2? Is it a Bayesian network?
> >
> > **A10:** We define the DAG in Line 133. The synthetic in parametric and non-parametric settings are generated following the DAG.
> >
> > **Q11:** The notion of selection “bias” is not rigorously described, which makes it hard to understand the theoretical results.
> >
> > **A11:**  We add the definition of selection bias in Section 2. The selection process introduced with the toy example in Line 45 leads to selection bias.

---

### Official Review · Reviewer_CZkW · 2024-11-02

**Soundness:** 3
**Presentation:** 1
**Contribution:** 3
**Rating:** 5
**Confidence:** 3

**Summary:**

This paper proposes GISL, a causal discovery method in the presence of selection bias and latent confounders.
The method leverages perturbations to break the asymmetry and discover the direction of causality between variables.
It further proves the identifiability of the causal dependencies and selection processes considering the absence of latent confounders
and ensures partial identifiability, which includes latent confounders.
Two variants of the methods GISB and GISL are evaluated empirically showing better performance over relevant baselines in synthetic scenarios.
In real gene expression data, GISL succeeds in learning the majority of known causal dependencies in between genes and selection processes.

**Strengths:**

The paper provides interesting and novel results. Among the contributions of the paper, I highlight the following strengths:
* GISL considers both selection bias and latent confounders and employs perturbations to discover the causal relations
* The discovered causal dependencies are guaranteed causal rather than spurious correlations.
* The authors provide theoretical guarantees for two scenarios, including one without latent confounders.
* Both variations GISB and GISL are evaluated empirically, validating the theoretical claims in practice.

**Weaknesses:**

The paper contains weaknesses in the writing, theory, and experiments.

**Writing.** The paper here requires improvement.

1. Some notions are not well introduced. For example, you should have explained in the background what is
identifiability/partial identifiability, faithfulness, and Markov condition.

2. The mathematical proofs are hard to follow. I recommend moving them to the technicalities in the appendix and instead bringing the corresponding figures in the main text.
Using the DAG figures you can explain the proof intuitively with examples.

3. Moreover, I have spotted mistakes in the grammar and syntax of the text:
* casual (ca-su-al) -> causal (in many places in the text)
* Line 119: Capitalize "We"
* Line 132: "The X =" -> "The data X ="
* Line 152: "In DAG, if" -> "In a DAG G, "
* Line 155: "If collider" -> "If the collider"
* Line 192: "1,3,5th lines" -> "lines 1, 3 and 5"
* Line 293: "If becomes all dependent patterns, keep it" -> I really don't understand what you write here, please fix it.
* Line 322: "IX → X → and IY → Y →" Maybe S missing in both arrows?

**Theory.** Regarding the theoretical results, Theorems 3.1, 4.1 do not quantify what "sufficiently many data" means. Moreover, it is unclear how you prove them by enumerating some possible causal configurations.
How are you sure that you have included all possibilities? Additionally, the number of required perturbations is not quantified.
Typically for biology experiments, each perturbation has cost, therefore a high number would not be efficient.

**Experiments.** The experiments provide validation for the theoretical claims but they lack in some aspects:
1. It is not determined how much data your method requires to perform reasonably.
What happens in the low data regime in comparison to the baselines?
2. You could have included a few more baselines like DCDI [1] and a method for gene regulatory networks [2].
3. Even though the result of the real experiment is valid, the DAG is very tiny (or did I miss something?).
There, you should have also compared against the baselines.

[1] Brouillard, Philippe, et al. "Differentiable causal discovery from interventional data." Advances in Neural Information Processing Systems 33 (2020): 21865-21877.

[2] Aibar, Sara, et al. "SCENIC: single-cell regulatory network inference and clustering." Nature methods 14.11 (2017): 1083-1086.

**Questions:**

* Line 155: Wouldn't this path create cycles? How is it possible?
* Line 190: What is $I$? I guess it is the surrogate. You should clearly define it and explain it.
* Line 348: Is it 30000 data in total for all 20 structures or for each of the 20? If it is for each then it is quite a large number of data.
* Line 366, "data to assist PC and GES in determining more edges": How do you implement this?
* In Table 1, only GISL is reported? Please mention which method is reported.

---

> ### Author Response · Authors · 2024-11-25
> **Responses to Reviewer CZkW**
>
> We thank your time, valuable feedback, and recommendations for improvement.
>
> **Q1:** Some notions are not well introduced.
>
> **A1:** Following your suggestions, we add definitions of the causal terms in Appendix A. Moreover, basic assumptions are cleared in Section 2 and Section 3.
>
> **Q2:** The mathematical proof is hard to follow.
>
> **A2:** Following your suggestions, we move the proof to the Appendix and use DAG figures to explain the proof intuitively.
>
> **Q3:** mistakes in the grammar and syntax of the text.
>
> **A3:** Thanks for pointing it out. We carefully check the paper and correct the mistakes.
>
> In Line 293, we add a detailed description of correcting rules in Section 4. Correcting rule 2 consider the case that when the collider is in the conditional set, which will lead to more dependence resulting in X and $I_{Y}$, Y and $I_{X}$ are always dependent. To tackle this case, we keep the previous results.
>
> In Line 322, it is not limited to the case that X and Y point to S. The $I_{X}$ → X → and $I_{Y}$ → Y → means that the CI patterns of the selection process are determined by the structure that the edge is out of X and Y as shown in Figure 12 (b). As long as the path between X and Y is always d-connected, we will get the CI patterns with the selection process. It is not limited to $I_{X}$ → X → S and $I_{Y}$ → Y →S.
>
> **Q4:** Theorems 3.1, and 4.1 do not quantify what "sufficiently many data" means.
>
> **A4:** In theory, we mentioned sufficient data only to say that our algorithm does not consider the error caused by insufficient data in theory, which makes the statement more precise. The constraint-based methods use the CI test, more data means more accuracy.
>
> **Q5:** How are you sure that you have included all possibilities?
>
> **A5:** The identifiability and partial identifiability are proved not by listing all possibilities case by case. The cases in Figure 10 are listed to help readers to understand the feasibility of the algorithm. In the proof of theorem 3.1, the proof by contradiction is utilized to verify that the algorithm is sound and complete under our assumption. In the proof of theorem 4.1, we discussed the feasibility of GISL with d-separated and d-connected (DAG-inducing) paths separately.
>
> **Q6:** The number of required perturbations is not quantified.
>
> **A6:** There are some misunderstandings about GISL. Our algorithm is not limited to the specific number of perturbations. For pair-wise causal discovery, if you would like to know the relation between any pair of genes, only the perturbation data of these two genes are required, which is also the priority of our algorithm.
>
> **Q7:** It is not determined how much data your method requires to perform reasonably.
>
> **A7:** The performance of all constraint-based causal discovery methods is based on the accuracy of the CI test, which is a hypothesis test. The performance of the CI test depends on several factors including the number of variables, and the complexity of the graph. The convergence rate and required sample size vary according to different CI test methods. The simple principle is that more data is more powerful. Based on the experimental experience, for example, 500 samples are enough for 50 variables for the Fisherz test.
>
> **Q8:** Line 155: Wouldn't this path create cycles? How is it possible?
>
> **A8:** It will not form a cycle. The inducing path is introduced in a fundamental paper on causal discovery [1]. The examples of DAG-inducing path like (b-d) in Figure 9. It will not create cycles. The DAG-inducing path only results in a d-connection between X and Y.
>
> [1] Zhang, Jiji. "On the completeness of orientation rules for causal discovery in the presence of latent confounders and selection bias." _Artificial Intelligence_ 172.16-17 (2008): 1873-1896.
>
> **Q9:**  Line 190: What is I? I guess it is the surrogate.
>
> **A9:** I is defined in Lines 134-135, which is the perturbation indicator, working as a surrogate variable in the graph. In Line 114, we also introduced I. We unify 'I' to the perturbation indicator for clarity.
>
> **Q10:** Line 348: Is it 30000 data in total for all 20 structures or for each of the 20?
>
> **A10:** 30000 sample size is for each structure before selection. After selection, it is around 10000.
>
> **Q11:** Line 366, "data to assist PC and GES in determining more edges": How do you implement this?
>
> **A11:** Under the setting of PC and GES, with the help of perturbation data, we follow the principle of asymmetry of causation, i.e., P(Y) != P(Y|do(X = X1)) if X causes Y.
>
>
> **Q12:** In Table 1, only GISL is reported? Please mention which method is reported.
>
> **A12:** Thanks for pointing it out. We verify the ability of GISB to identify the selection process in Table 1. In essence, GISB and GISL model observational and interventional data in the same way. One method is enough to verify it.

---

> > ### Comment · Reviewer_CZkW · 2024-11-26
> > **Thanks for your response and changes to the paper**
> >
> > I thank their authors for their clarifications and changes in the paper.
> >
> > I want to point out that partial identifiability and identifiability are still not concretely defined, but rather they are implicitly shown with two theorems. You should use definitions to explain them. Also, I still believe that this work would benefit from a theoretical result quantifying the number of data with respect to the method's performance. (how many data ? infinite, exponential, or something else?)
> >
> > The authors have partially addressed my concerns. I will thus maintain my score.
> >
> > In particular, the following points were not addressed:
> > * What happens in the low data regime in comparison to the baselines?
> > * You could have included a few more baselines like DCDI [1] and a method for gene regulatory networks [2].
> > * Even though the result of the real experiment is valid, the DAG is very tiny (or did I miss something?). There, you should have also compared against the baselines.

---

> > > ### Author Response · Authors · 2024-12-01
> > >
> > > Thanks for your commonts.
> > >
> > > We apologize for the late reply.
> > >
> > > **Q1** What happens in the low data regime in comparison to the baselines?
> > >
> > > **A1** We believe the low-data setting is already addressed in our experiments. During the generation of synthetic datasets, the selection processes reduce the data to only a few thousand samples. Furthermore, additional samples are removed after perturbation as the selection processes remain active. As a result, the final sample size for the perturbation data ranges from 100 to a few thousand.
> > >
> > > **Q2** Baselines.
> > >
> > > **A2** Following your suggestions, the computational methods for GRNI are included as baselines, and the experimental results are presented in Table 2. When computational methods encounter selection bias, they even fail to find good dependencies. We carefully analyzed the results and observed that they tend to produce an almost fully connected graph. This occurs because selection bias affects the distribution of all nodes along the path. The results of DCDI compared with GISL in the general case are as follows:
> > > | Method     | ACC   | F1  |  Recall   | SHD  |
> > > |:----------|:--------:|--------:|--------:|--------:|
> > > | GISL  |  0.88 &plusmn; 0.04  | 0.84 &plusmn;  0.05   |  081 &plusmn;  0.06| 0.75 &plusmn; 0.98|
> > > | DCDI   | 0.2 | 0.33   |   1 |   10 |
> > >
> > > The DCDI utilized interventional data, but it still cannot handle selection bias. The reason is the same as the analysis before. Extensive distribution change can be detected on the nodes that are on the path of selection pairs. As a result, the output is also an almost fully connected graph.
> > >
> > > **Q3** The DAG is tiny.
> > >
> > > **A3** Our proposed GISL can handle 5012 or more genes. We show the graph of perturbed genes in Figure 16. The GISL has two stages. First, the skeleton is discovered among 5012 genes. Then, perturbation data are used for identifying the structure of node pairs for perturbed genes. However, the DCDI and other computational baselines can not handle 5012 genes due to the high requirement of memory. For example, the error of DCDI is "MemoryError: Unable to allocate 91.4 TiB for an array with shape (1000000, 5012, 5012) and data type float32". We reported the results of computational methods in Figure 17 if applicable.

---

### Official Review · Reviewer_6t7r · 2024-11-03

**Soundness:** 2
**Presentation:** 2
**Contribution:** 2
**Rating:** 3
**Confidence:** 4

**Summary:**

This paper addresses the problem of selection bias and latent confounders in Gene Regulatory Network Inference (GRNI) by proposing the GISL algorithm, which uses observational and gene perturbation data to identify causal relationships, selection processes, and latent confounders. The authors highlight that these factors are prevalent in real biological data but are often overlooked by existing methods. The study is well-motivated, with both theoretical and practical significance. Through theoretical analysis and experimental validation, the authors demonstrate the effectiveness of GISL in identifying causal structures in gene regulatory networks.

**Strengths:**

The idea of considering selection inclusion and latent confounders when inferring gene regulatory networks is of interest especially in the bioinformatic domain.
The author describes the fundamental algorithm well, and they seem to give all relevant information to understand and reproduce their algorithm.

**Weaknesses:**

Some definitions and descriptions in the text are vague or misleading.
The authors did not compare their method with latest state-of-the-art methods.
The data scale in the experimental section is too small, leading to an insufficient evaluation.

**Questions:**

1. Some definitions and descriptions in the text are vague or misleading. For example, in Definition 2.3, the phrase "Toy examples are shown in 9" is unclear. In the description of Figure 9, they mention "Definition 2.5" which is not provided in the main text, they may refer to Definition 2.3. Such inconsistencies in terminology arise multiple times, making the text difficult to read. Additionally, the phrase "relative to" in Definition 2.3 lacks clarity, and the three conditions listed are overly complex and challenging to interpret. The authors are encouraged to standardize their terminology definitions to enhance consistency and clarity.

2. Taking the GISB algorithm (Algorithm 1) as an example, the description of the algorithm is somewhat incomplete:

Step 3: It lacks specific details on the marginal and conditional independence tests, including the methods used for testing and the criteria for decision-making.

Step 4: It does not clarify how to select gene subsets along the path for conditioning or how to update the results.

Unclear Symbols and Variables: The symbol "I" is referred to as a "surrogate variable" in the algorithm but is labeled as a "perturbation indicator" in the proof. It would be beneficial to unify and clarify its meaning. Additionally, the variable introduced in Step 4 is not adequately explained beforehand.

3. The experiments do not include comparisons with methods developed after 2023. Given the rapid advancements in causal inference techniques, many recent algorithms are better equipped to handle nonlinear dependencies and selection bias. It is recommended to benchmark GISL against the latest methods, particularly those utilizing deep learning models for causal discovery, to further validate its effectiveness and novelty.

4. The experiments utilize a limited range of node counts (5 to 9 nodes). Expanding the study to include larger-scale networks would enhance the generalizability of the findings and further validate the applicability of GISL in more extensive networks.

---

> ### Author Response · Authors · 2024-11-25
> **Responses to Reviewer 6t7r**
>
> Thank you for your time and suggestions.
>
> **Q1:** Some definitions and descriptions in the text are vague or misleading.
>
> **A1:** Thanks for pointing it out.  We corrected the unclear part in the description of Figure 9. For Definition 2.3, we follow the paper [1], which introduces the inducing path. We know it is a little complex to understand, but it is commonly used in causal communities.
>
> [1] Zhang, Jiji. "On the completeness of orientation rules for causal discovery in the presence of latent confounders and selection bias." _Artificial Intelligence_ 172.16-17 (2008): 1873-1896.
>
> **Q2:** Taking the GISB algorithm (Algorithm 1) as an example, the description of the algorithm is somewhat incomplete:
>
> **A2:** We add more descriptions for GISB and GISL in the Motivation and Discussion part. The description of concepts is added in Appendix A. More details of GISB can be found in Appendix (Algorithm 3). In Figure 8, we have shown the differences in marginal and conditional Independent tests, and it is utilized starting from Line 729. In Lines 735-745, we introduced the details about how to select gene subsets and how to update results.  We unify the description of the symbol 'I" with the perturbation indicator, which works as a surrogate variable in the graph.
>
> **Q3:** More strong baselines are needed for comparison.
>
> **A3:** As we know, besides FCI, this is the first algorithm that can do causal discovery in the presence of latent confounders and selection bias in a general setting. I agree that many methods can handle non-linear or causal inference under the selection process. However, most of them do causal inference under selection by knowing the structure. Their setting is only focused on one part of the problem, it is not the same general setting as ours. If there are methods that can handle the problem we introduced, please let us know. Thanks!
>
> **Q4:** If the algorithm extends to large-scale networks?
>
> **A4:** The GISL algorithm has two stages, stage 1 is skeleton discovery on observational data, which may have scalability problems. However, this is not a problem, FGES and parallel PC can eliminate the concern. The running time of FGES on real-world (5045 genes) and synthetic (5806 genes) datasets is 195 minutes and 335 minutes respectively. The second stage is pair-wised causal discovery on observational and perturbation data, which does not suffer the heavy computation problem. Only the perturbation data for the paired genes whose relationship you wish to determine is required.

---

> > ### Author Response · Authors · 2024-12-01
> >
> > Apologies for the late update. The experimental results on 5012 genes are reported in Figure 16. Please kindly refer to the Appendix H for details. If any concerns please let us know. Thanks!

---

### Official Review · Reviewer_YBTw · 2024-11-03

**Soundness:** 4
**Presentation:** 3
**Contribution:** 2
**Rating:** 5
**Confidence:** 5

**Summary:**

This paper mainly addresses the inference of gene regulatory networks (GRNs) based on single cell perturb-seq data. Traditional Single cell RNA seq data provides a count table of transcribed RNAs so it offers a snapshot of gene expression profile of the cell state. With the help of CRISPER editing, we can knock down/off a single gene and then observe the expression of all the other genes in this condition.

While people may think we can recover the causal dependencies by directly comparing the difference between knock off and wild type in perturb seq, the authors of this paper point out this is not the case. With evidence, they argue that "dependencies can be generated in three ways: through causality, latent confounders, or selection bias". So the relationships we observe could totally be false positive. Later in the paper, they provide solutions/partial solutions for these two errors.

In the case with selection bias but without latent confounder, the author argue that the dependencies coming from selection bias would be symmetric while the true causal relationship should be asymmetric (based on the DAG assumption). Essentially, they prove that with the perturbation data, the causal structure could be identified by testing the marginal and conditional independence.

In the case that comes with both selection bias and latent confounder, they argue that the dependencies from latent confounder should also be symmetric. However, since the introduction of latent confounder also increases the freedom, the causal structure could be partially identified.

This paper is backed up by experiments on both synthetic and real world data.

**Strengths:**

For the field of GRN inference, the discussion in this paper are not commonly discussed or well understood by the community. I think this paper do provide some valuable insights from the perspective of theoretical causal inference/discovery.

The introduction of this paper is well motivated and attractive. The theoretical piece of this paper is very nicely written, clear and technically sound. The proposed method is novel and make sense. The flow of the paper is also sequential and well planned. The experiment on synthetic data is well designed and the results supports the main claims of this paper.

**Weaknesses:**

While I do enjoy reading this submission and I really like the concepts and awareness proposed in this paper, I do have a significant concern in terms of the actual usability of this method in real world and I think the flaw is in fact in contradiction of what the authors are trying to achieve.

1. One of the two main motivations of this study is to address the potential existence of latent confounder. The real world use case that the author provided is the existence of non-coding RNA, which is true. However, before we even come to non-coding RNA, can we make sure that we can at least have the capacity to model all protein coding genes? Apparently, this is not possible with the proposed method. We know that human have 20,000 protein coding genes and even yeasts have 6,000 genes. The PC method that the author used in this study is not really scalable at this scale and the added iterative updating mechanism of the proposed algorithm only makes the runtime worse. In fact, in the real world experiments of this study, only **5** genes are considered. In the synthetic example, the largest net has roughly 20 nodes. It means that to use this algorithm, you can only use a tiny slice (0.025 - 0.1%) of all the available data. So some of the genes that you removed ultimately become the "latent confounders" while they are not actually "latent"... That says, you are literally creating problems to solve for no good reasons... This is my biggest concern with this paper. Please provide a justification on that. You may consider replacing PC with more scalable methods (not sure if it's possible for the time frame) or discuss the case when heavy gene filtration is necessary and valid.

2. In connection with point 1, please provide a discussion on the runtime and provide clock time for the synthetic and real world experiment.

3. Baseline methods in this study includes PC, GES, and GIES. They are well-known causal structural learning methods. However, since we are talking about GRN inference here, do you want to include some common GRN inference methods, such as GENIE3, Grnboost or DCI (as you mentioned it in the introduction)?

4. In the real world experiment, can we have a figure of the results on the same data from PC without the updates in GISL?

5. The literature review on GRN inference on page 2 needs some improvements. Right now, it's mostly focusing on causal discovery. There are only 5 references that are actually talking about GRN and 2 of them are not GRN inference method paper. Levine's 2005 PNAS paper talks about GRN but it has nothing to do with GRN inference based on transcriptomics data. Pratapa's BEELINE is a standard benchmarking dataset in this field but it's not a method. The other 3 paper are only briefly mentioned in one sentence.

**Questions:**

Again, my biggest concern and question is mentioned in the weakness section.

Also, here is a general question that I have: many efforts in this method is to turn undirected edges predicted by PC to directional. However, we know that some genes are working in groups and they are just co-expressed as a group. A good example would be those histone genes. In gene network, you will see they form big hair balls. You mentioned that symmetry is a sign for either selection bias or latent cofounder. Do you think this kind of co-expression is a pattern for selection bias or something else? Do they really need to be directional?

---

> ### Author Response · Authors · 2024-11-25
> **Reponses to Reviewer YBTw**
>
> We appreciate your time, valuable feedback, and recommendations for improvement. Thank you so much for enjoying our work.
>
> **Q1:**  Feasibility of GISL algorithm in processing 6000 genes in real biological data.
>
> **A1:** Our algorithm can handle 6000 or more genes. First, let me clarify some details. There are two stages in the algorithm. The first stage is skeleton discovery on observational data, which is the point you are concerned about. This part is not limited to the traditional PC method. The goal of skeleton discovery is to improve the efficacy of the Conditional Independent test by reducing the size of conditional sets. Paralleled methods like parallel PC and FGES can be applied to tackle your concern. Thanks for pointing it out, we should clarify this part in the paper.  The second stage is pair-wised causal discovery on observational and perturbation data, which does not suffer the heavy computation problem. Only the perturbation data for the paired genes whose relationship you wish to determine is required.
>
> We test FGES on both synthetic and real-world data. The real-world data is the Norman dataset, with 5045 genes. The running time is 11644.288349866867 seconds, i.e. **195** minutes. The synthetic dataset is built following the structure generated by the Erdos–Renyi model with 5806 nodes. The running time of FGES on it is 20061.518082618713 seconds, i.e. **335** minutes. For generating a more dense skeleton with 5045 genes, it takes 1570 minutes.
>
> **Q2:** The running time of the algorithm on real-world and synthetic datasets.
>
> **A2:** As we mentioned in **A1**, the running time of FGES on real-world and synthetic datasets is 195 minutes and 335 minutes. The running time is also highly related to the complexity of the graph as we tested.
>
> **Q3:** If baselines should consider computational method?
>
> **A3:** We compared GISB and strong computational baselines (PIDC, PPOCR) as shown in Table 2 to illustrate why we do not consider them. In our view, the reason why we do not consider computational methods is that the output is the undirected graph (dependence only) instead of DAG, which is not what we want for GRNI. As selection will affect the distribution of all nodes on one path, computational methods result in an almost fully connected graph. The failure of computational methods indicates that they can not handle selection bias.
>
> **Q4:** The PC on real-world data.
>
> **A4:** The result of the PC is the same as the skeleton with black edges shown in Figure 7, which fails to find the direction. We would like to show you the result on a large graph with 5000 genes. As time is limited and massive dropouts need to be taken into account, we are still waiting for the one. At the same time, we are eager to have a discussion with you. We will update it for you asap.
>
> **Q5:** Literature review on computational GRN baselines needs improvements.
>
> **A5:**  Thanks for your suggestion. We have improved the references of the computational methods in Section 1. In the definition of Gene regulatory network inference, regulatory relation is defined as a causal relation. We rethink it from a causal view. As we discussed the performance in **A3**, computational methods can not handle spurious relations introduced by selection processes and latent confounders. That's why we focus more on causal.
>
> **Q6:** Gene Group co-expression.
>
> **A6:** Good example! For the genes that coexpress as a group, there must be some constraints or latent confounders to form the coexpression pattern (dependence). With our algorithm, the reason why grouping will be figured out by detecting latent confounders and selection processes. Moreover, directional regulatory relations are needed. For each pair of genes, no matter whether in groups or out of groups, discovering directional edges is necessary to know how one gene affects the other one, which is meaningful in utilizing this relation for disease.

---

> > ### Comment · Reviewer_YBTw · 2024-12-01
> >
> > Thanks for the updates and your answers to my questions. Also sorry for the delay on my end.
> >
> > 1. It's great to know that GISL can at least scale to 6000 or more genes. However, how does it perform at large scale, especially when it's on real data instead of synthetic?
> > 3. Also, thanks for adding table 2. However, as we all know there is a very large gap between synthetic data and real data in this field. Given that PIDC is a strong baseline and it has already generated many useful biological insights, I don't understand how its performance could be so low while the performance of GISL is so high. I wonder if it's because the evaluation was done on synthetic data.

---

> > > ### Author Response · Authors · 2024-12-01
> > >
> > > Thanks for your comments.
> > >
> > > 1. Let us introduce the procedure step by step. We conduct experiments on real-world datasets, e.g., Dixit which includes 5012 genes. Step 1, the skeleton of genes is discovered from observational data (5012 genes). This can be done by FGES, which can handle a graph with millions of nodes. The running time we updated is for Step 1. Step 2, the structure of any pair of genes is identified with the help of perturbation data. Here any pair of genes means the perturbed genes not all genes. Finally, the structure of perturbed genes will be identified. Genes without corresponding perturbation data can not be identified. We update the results of Dixit in Figure 16. The practical meaning of GISL is discussed in the last sentence of Section 5.
> > >
> > > 2. The synthetic data includes the selection process. In the view of PIDC, if only dependencies are considered, then it will output an almost fully connected graph. This is because each selection works on the whole path, the distribution of all nodes on the path will change, which results in lots of spurious dependencies. No matter the distribution change or dependencies the methods rely on, it does not work for the selection process. This is why computational and current causal discovery methods fail in GRNI. We tried PIDC, but it suffers the low-memory problem on a large-scale graph. We update PPCOR on Dixit as shown in Figure 17.

---

### Meta-Review · Area_Chair_9cTo · 2024-12-17

**Metareview:**

This paper addresses the problem of selection bias and latent confounders in Gene Regulatory Network (GRN) inference. The paper claims that these effects have been overlooked and perturbations can yield insights.

The reviews indicated that there were strengths in the theoretical and synthetic data experimental aspects of the paper.

The reviews also noted several weaknesses. There were concerns about scalability, comparisons to existing methods, and incorporation of relevant literature. There were also concerns about vagueness of the claims and the ability of these claims to rest on the evidence presented.

Given these concerns the paper is not above the bar for acceptance at this time, but the modifications suggested by the reviewers and partially addressed in the author rebuttal will certainly improve the paper.

**Additional Comments On Reviewer Discussion:**

The reviewers and authors engaged in an extensive discussion on the merits of the paper. The author rebuttal addresses some of these concerns with substantive new experiments or data.

The totality of the reviewer discussion is considered in the decision on the submission. Of course, some reviewers will be closer to the specific field of study while others are slightly more distant. And, it's important to collect this diversity of perspectives in order to make a good decision for the entire community. The reviewers and authors all engaged in productive dialogue on this submission to understand the work and presents its strengths and weaknesses.

---

### Decision · Program_Chairs · 2025-01-22

Reject